



# Sensitivity analysis of a data-driven model of ocean temperature

Rachel Furner[1,2], Peter Haynes[1], Dave Munday[2], Brooks Paige[3], Daniel C. Jones[2], and
Emily Shuckburgh[1]

[1]University of Cambridge, Cambridge, UK
[2]British Antarctic Survey, Cambridge, UK
[3]University College London, London, UK

**Correspondence:** Rachel Furner (rachel.furner@maths.cam.ac.uk)

**Abstract.** There has been much recent interest in developing data-driven models for weather and climate predictions. These have shown reasonable success in modelling atmospheric dynamics over short time scales, however there are open questions regarding the sensitivity and robustness of these models. Using model interpretation techniques to better understand how data-driven models are making predictions is critical to developing trust in these alternative prediction systems. We develop and interpret a simple regression model of ocean temperature evolution, to improve understanding of whether data-driven models are capable of learning the complex underlying dynamics of the systems being modelled.

We investigate model sensitivity in a variety of ways and find that the simple regression model analysed here behaves in ways which are, for the most part, in line with our knowledge of the ocean system being modelled. Specifically we see that the regressor heavily bases its forecasts on, and is dependent on, variables which we know are key to the physical dynamics inherent in the system, such as the currents and density. By contrast, inputs to the regressor which have limited direct dynamic impact, such as location, are not heavily used by the regressor. We also find that the regression model requires non-linear interactions between inputs in order to show any meaningful predictive skill — in line with out knowledge of the highly nonlinear dynamics of the ocean. Further sensitivity analysis is carried out to interpret that the ways in which certain variables are used by the regression model. Results here are again mostly in line with our physical knowledge of the system, for example, we see that information about the vertical profile of the water column reduces errors in areas associated with convective activity, and information about the currents is used by the regressor to reduce errors in regions dominated by advective processes.

Our results show that even a simple regression model is capable of 'learning' much of the physical dynamics inherent in the ocean system being modelled, which gives promise for the sensitivity and generalisability of data-driven models more generally.

## 1 Introduction

**Data-driven models for weather and climate**

Applications of Machine Learning (ML) in weather and climate modelling, of both the ocean and atmosphere, have seen a huge rise in recent years. Traditionally, weather and climate prediction relies on physics-based computational models of the earth system, hereafter referred to as simulators or General Circulation Models (GCMs). Recently a number of papers have





focused on creating statistical/data-driven models for a variety of physical systems (Miyanawala and Jaimana, 2017; Pathak et al., 2018; Breen et al., 2020). These show the ability for statistics and ML to compliment existing methods for predicting the evolution of a range of physical systems.

Lorenz models (Lorenz, 2006) are often used as a simple analogous system for weather and climate models as they have similar properties albeit a considerably simplified way. Many data-driven models of the Lorenz equations have been developed and assessed, i.e. Dueben and Bauer (2018), Chattopadhyay et al. (2019), Doan et al. (2019), Scher and Messori (2019b). These results show that data-driven methods can capture the chaotic dynamics of the Lorenz system, and make skilled, short-term forecasts. A number of papers (Dueben and Bauer (2018), Scher (2018), Scher and Messori (2019a) Weyn et al. (2019), Arcomano et al. (2020) and Rasp and Thuerey (2021)) go further and apply statistical and ML methods to simple weather prediction applications, using a variety of model architectures, and training on both observational data and GCM output. Rasp et al. (2020) looks to standardise and formalise comparison of these methods, proposing a common dataset and test experiments to enable assessment of methods for predicting the short term evolution of the atmosphere within a common framework. The development of this field provides great promise for weather and climate prediction, with the demonstration of skilful forecasts, which could one day be used to provide efficient operational forecasts to complement existing physics-based GCMs.

**Interpretable ML**

Using data-driven methods in place of physics-based GCMs raises questions about how these models are making their predictions, and the reliability and generalisability of these models. GCMs are based on known physics, meaning a single model can be used to reliably predict a variety of regimes. Data-driven models are instead dependent upon the data used during training, and the patterns learned by the model. The ability for a data-driven model to generalise, that is to make skilful predictions for data which differs in some way to the data seen in the training set, depends on how the predictions are being made. If statistically robust patterns or links are found that hold well within the training data, but which ultimately have no physical basis, then we would not necessarily expect these models to perform well on data outside of the training set. For new examples, which bear little similarity to that seen in the training data and which are not close to any training examples in feature space, i.e. extreme events not included in the training set, any non-physical patterns that were learned are unlikely to hold and the model will not necessarily be expected to perform well. By contrast, if the performance over the training data is skilful because the model is learning meaningful physical links between the input and output variables, then we would expect for the model to perform well for any data which exhibits these same physics, irrespective of the similarity to training samples. If data-driven methods for predicting weather and climate systems are able to learn the underlying dynamics of the system, rather than statistically valid but non-physical patterns between inputs and outputs, we have increased confidence that these systems can be usable for a wide variety of applications.

A number of techniques exist to understand the sensitivity of data-driven models, and to interpret how they are making their predictions, giving insight into their generalisability and reliability (McGovern et al., 2019; Molnar et al., 2020). These techniques seek to help us understand not just whether a model is getting the right results, but if the models are getting the right results for (what we consider to be) the right reasons, i.e. by learning meaningful physically consistent patterns. There are



60 a number of Model Interpretation and Visualisation (MIV) techniques, which focus on different elements of interpretability. Methods look at both identifying which features are important to a model (i.e. sequential search, impurity importance) and assessing how certain features are used by the model (i.e. partial dependence plots, saliency maps). These methods seek to answer subtly different questions about how a model is working, and so it is common to use a variety of model interpretation techniques in parallel. Techniques which assess feature importance highlight which features are fundamental to the forecast,

65 but not how these are being used. By contrast, methods which look at how features impact the forecast do not indicate the relative importance of the these features for the predictions. As data-driven methods become more commonly used in weather and climate applications, so does analysis into the interpretability of these models, i.e. Mcgovern et al. (2020) and Rasp and Thuerey (2021).

## A regression model of ocean temperature

The studies mentioned above focus on atmospheric evolution, here we focus on oceanic evolution. We develop a data-driven model to predict change in ocean temperature over a day, based on data from a GCM of the ocean and then interpret this model through a variety of methods. The underlying physics explaining the dynamics of the Earth system is consistent across the atmosphere and ocean. Whilst there are many differences between atmospheric and ocean dynamics, for example the temporal and spatial scales of interest, and compressibility of the fluid, these systems are driven by similar physics. As such, the skill shown in using data-driven methods for predictions of the atmosphere suggest that these same methods could provide skilful predictions for the evolution of the ocean.

We apply model interpretation techniques to our data-driven model to try to understand what the model is 'learning' and how the predictions are being made, and compare this with our prior knowledge of the ocean dynamics. We analyse the sensitivity of the regressor to its input variables, firstly through direct analysis of the coefficients of the resultant model to show which variables are heavily used in the forecasts, and secondly through withholding experiments to indicate which variables are necessary for producing skilful forecasts. Lastly we further analyse some of the withholding experiments to infer *how* some of these key variables are contributing to the predictions.

Section 2 discusses methods: the GCM we use to create our training and validation dataset; the regressor we develop; and the sensitivity analysis we perform. Section 3 discusses the skill of the developed regressor. Section 4 explores the sensitivity of the regressor to its inputs. The results and their implications are discussed in Sect. 5.

## 2   Methods

### 2.1   Simulator generated dataset

#### 2.1.1   Simulator configuration

Our training and validation data comes from running the MIT General Circulation Model (MITgcm). This is a physically based model capable of simulating the ocean or atmosphere due to isomorphisms in the governing equations (Marshall et al., 1997a,





b). Specifically we use a 2° sector configuration following Munday et al. (2013) to simulate ocean dynamics. This configuration features a single ocean basin, with limited topography, simplified coastlines, and constant idealised forcing. This has been used in a number of idealised simulations of Southern Ocean processes and their impacts on the global circulation (Munday et al., 2013, 2014). This configuration, whilst relatively simple, captures the fundamental dynamics of the ocean, including a realistic overturning circulation.

The domain runs from -60°S to 60°N, and is just over 20°wide in longitude. The domain is bounded by land along the northern (and southern) edge and a strip of land runs along the eastern (and western) boundary from 60°N to 40°S (see Fig. 1a). Below this, in the southern-most 20°, the simulator has a periodic boundary condition, allowing flow which exits to the east (west) to return to the domain at the western (eastern) boundary. The domain has flat-bottom bathymetry of 5000m over most of the domain, with a 2° region of 2500m depth at the southern-most 20°of the eastern edge (i.e. the spit of land forming the eastern boundary continues to the southern boundary as a 2500m high sub surface ridge).

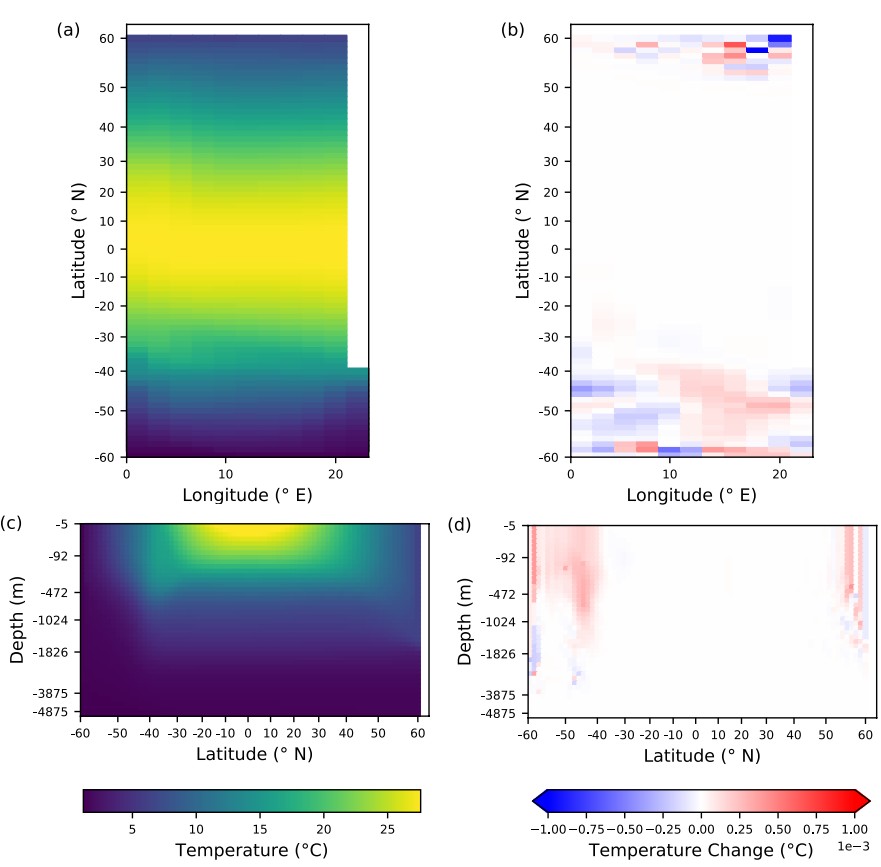

**Figure 1.** Left: Plot of simulator temperature (°C) at 25m below the surface (top) and at 13° E (bottom), for one particular day, and Right: the change in temperature between this day and the next. Note the depth axis is scaled to give each GCM grid cell equal spacing.

The simulator shows a realistic temperature distribution with warm surface water near the equator, and cooler water near the poles and in the deep ocean. Temperature changes are largest in the very north of the domain and throughout the southern region.





The simulator is run with 42 (unevenly spaced) depth levels, following a Z-coordinate, and has 11 cells in the X (longitudinal) direction, and 78 cells in the Y (latitudinal) direction. The grid spacing is 2° in the Y direction, with the X spacing scaled by

the cosine of latitude to maintain approximately square grid boxes (this means grid cells close to the poles are about a factor of 4 smaller than those near the equator, but all cells remain approximately square). The simulator has a 12 hour time step (two steps per day), with fields output daily. We focus on daily-mean outputs, rather than the instantaneous state.

At 2° resolution the simulator is not eddy resolving, but uses the Gent–McWilliams parameterisation (Gent and Mcwilliams, 1990) to represent the effects of ocean eddy transport. We ran the simulator with a strong surface restoring condition on

Temperature and Salinity — thus fixing the surface density. We apply simple jet-like wind forcing, constant in time, with a sinusoidal distribution between 60°S and 30°S, with a peak wind stress value of 0.2 Nm$^{-2}$ at 45°s.

### 2.1.2   Ocean dynamics

We are interested in predicting the change in temperature between two successive daily mean values. Figure 1 shows the daily mean temperature for a given day, along with the one day temperature change, for cross sections at 25m depth and 13°

longitude.

From Fig. 1 we can see the simulator represents a realistic temperature distribution, with warm water at the surface near the equator and cooler water nearer the poles and at depth. The largest changes in temperature over a day are located in the south of the domain, and in a small region in the very north. These changes result predominantly from the local vertical activity associated with the Meridional Overturning Circulation (MOC) — a wind and density driven circulation that is characterised

by water sinking in the north, travelling southward at depth, and then upwelling in the south, where it splits in two, with some water returning northward near the surface, and some re-sinking to the south and returning north at depth (Talley, 2013; Rintoul, 2018). This circulation occurs on timescales of hundreds to thousands of years, i.e. this is the time taken for water parcels to complete one revolution, however the local vertical movements associated with it drive the largest daily temperature changes seen in the far north and south of this domain. The MOC reflects the density profile, which itself arises from the surface forcing

(the restoring term on temperature and salinity) and the wind forcing.

Further details of the model dynamics, in particular assessment of the contribution of different processes to temperature change in the model can be found in Appendix A.

### 2.1.3   Training and validation datasets

Input and output data for training and validating the regressor comes from a 70 year run of this simulator. The first 50 years

of the run are discarded, as this includes a period where the model is dynamically adjusting to its initial conditions which may be physically inconsistent. During this period the evolution of the simulator is driven by this adjustment, rather than the more realistic ocean dynamics which we are interested in, hence we exclude this data. This leaves 20 years of data which is used for training and validating the model. As the GCM sees constant wind forcing and a consistent restoring of surface temperature and salinity, if left to run for long enough (thousands of years) the system would reach a quasi-steady state, however the 20

year period used here is prior to the model reaching this quasi-steady state.





As the data is highly auto-correlated we sub-sample in time to remove some of the co-dependent nature of the training data. There are also computational constraints limiting the total size of our dataset. This leads us to choose a subsampling rate of every 200th day. This dataset is then split into training, validation and test data with a split of 70-20-10. The data is systematically split temporally, so the first 70% of samples are used as training data etc, meaning each dataset contains data from different temporal sections of the run, maximising independence across the datasets. For every 200th day, we take all grid points from throughout the model interior (i.e. we exclude points next the land, and points at the surface and seabed). We do not subsample in space, as the domain is reasonably small and the dynamics varies considerably across it, meaning subsampling in space can lead to some dynamic regimes being entirely missing from the dataset. This gives us approximately 650 000 training samples, 200 000 validation samples, and 100 000 test samples.

## 2.2 Regression model

We develop a linear regression model to predict the daily mean change in ocean temperature for any single grid cell, based on variables at surrounding locations at the current time step. The regressor is defined such that it outputs temperature change at a single grid cell rather than predicting for the entire domain, but the cell being predicted can be any of the cells in the domain interior — the regressor is not limited to predicting for a specific location.

Equation 1 shows the formulation of the regressor.

$$\hat{y} = \sum_{i=1}^{N_f} \beta_i x_i + \sum_{\substack{i,j=1 \\ i<j}}^{N_f} \gamma_{i,j} x_i x_j \tag{1}$$

Here $\hat{y}$ is the output from the regressor — an estimate of the change in daily mean temperature over a day for the grid cell being predicted. This is calculated as the mean temperature at the next day ($t+1$) minus the mean temperature at the present day ($t$). $N_f$ is the number of input features used in the model. $\beta_i$ and $\gamma_{i,j}$ are the weights of the regressor which are learnt during the training phase. The $x$'s are the input features being used to make the predictions.

Input variables are temperature, salinity, U (East-West) & V (North-South) current components, density, U, V & W (vertical) components of the Gent-McWilliams (GM) bolus velocities, sea surface height (SSH), latitude, longitude and depth. The GM bolus velocities are a parameterisation of the velocities resulting from ocean eddies and are used in the GM scheme to calculate the advective effects of eddy action on tracers. For 3d variables (temperature, salinity, current components, density, and GM bolus velocity components) input features are taken from a 3x3x3 stencil of grid cells, where the centre cell is the point for which we are predicting, giving 27 input features for each variable. For SSH, which is a 2d variable, the values over a 2-d (3x3) stencil of surrounding locations are included, giving a further 9 features. Lastly, the location information (latitude, longitude and depth) at only the point we are predicting is included, giving the final 3 input features. All temporally changing variables are taken at the present-day (t). In total this gives $N_f = 228$ features, represented by the first term in Eq. 1.

We also include second order pairwise polynomial terms, in order to capture a limited amount of non-linear behaviour through interaction between terms. This means that as well as the above inputs, we have multiplicative pairs of features, represented by the second term in Eq. 1. Note we include second order interactions between different features, but not squared



terms, as we are interested in representing the interaction between different features through this term. This gives 26 016 input terms in total.

The model design means that all physical ocean variables at surrounding points are included in the prediction, as these are likely to impact the change in temperature at the central point. Geographic inhomogeneity is accounted for through inclusion of the location information. Further, the combination of this geographic inhomogeneity with physical ocean variables is included to a limited extent through some of the multiplicative terms in Eq. 1 (those terms which are a combination of latitude, longitude or depth with a physical variable input). Lastly, the non-linear interaction between physical ocean variables is also included to

a limited extent through the remainder of the multiplicative terms. All input variables are normalised prior to fitting the model by subtracting their mean and dividing by their standard deviation.

### 2.2.1   Limitations of the model

It should be noted that the model is a simple regressor, to allow for easy analysis of sensitivity. This however limits how accurately the model can fit the data, and how well it can represent the underlying system. In particular, we know the ocean to

be highly non-linear, but allow only second order polynomial terms in the regressor, restricting the level to which it can capture the true dynamics.

The regressor here takes input data from only immediately surrounding grid cells, meaning it has no information about what is happening in the wider domain. This potentially prevents the regressor from making predictions far ahead, when the wider ocean state has more influence, but for the short time steps being forecast here (one day), this local information is expected to

be sufficient. Indeed, here we are making predictions at time steps only double that used in the GCM — where the change at each cell is based predominantly on the state of only immediately surrounding cells.

We note that many existing papers looking at data-driven forecast systems focus on developing methods which can be applied iteratively to provide an alternative forecast system able to predict an arbitrary number of time steps ahead. We emphasise that the model described here would not, in its current form, be usable to produce an iterative forecast in this same way. There are

two main reasons for this. Firstly, this model is unable to forecast near the boundary as it requires a full set of neighbouring input points. Secondly, this model requires a wider set of inputs than the outputs it produces. This means that if using this iteratively as an alternative forecast system to a GCM, if a full set of initial conditions were given, we would still require some means of generating variables other than temperature, in order to provide the full set of inputs to the regressor for future time steps. Our interest here is not in deriving a data-driven analogue of the MITgcm simulation which might one day be used in

place of the original simulator, but simply in assessing the sensitivity of this data-driven model to different variables. The set up described here best allows us to assess the dependencies and sensitivities of a simple data-driven model to various inputs, and thus provides insight into the general ability of data-driven methods to learn the underlying dynamics of weather systems.

### 2.2.2   Training the regressor

The model is trained by minimising least squares errors with ridge regularisation (Hoerl and Kennard, 1970). Training a stan-

dard least squares model amounts to finding values of the coefficients ($\beta_i$ & $\gamma_{i,j}$) which minimise the squared difference





between the regression model predictions and the actual outputs taken from the GCM over the training dataset. In any application of a regression model it is expected that the model will be used on data other than that used in the training of the model. To ensure the model performs well on unseen data, we want to ensure that that model learns the general pattern of the data, rather than specifically fitting to every point in the training data. This is particularly important where datasets are known to contain

noise, as here fitting the data exactly would mean 'learning' the noisy representation of the system that the data portrays, rather than learning the underlying system itself. Regularisation techniques are applied to avoid the problem of overfitting (of matching the training data exactly) and work to limit the level at which the model can fit the data, ensuring the model can generalise well — i.e. it still performs well on new unseen data which shares the same underlying dynamics. Ridge regression is one such regularisation method, which works by minimising the size of the coefficients as well as the model errors. When using ridge

regression an additional term is added to the loss function, so the training now focuses on minimising a combination of the square errors and the sum of the magnitude of the $\beta_i$ & $\gamma_{i,j}$ values, with $\alpha$ acting as a tuning parameter determining the balance between these two terms.

We use a very small value of $\alpha = 0.001$. This was found through cross validation with values of $\alpha$ ranging from 0.001 to 30. With larger values the regressor performed poorly, particularly when predicting larger temperature changes. Given the

dataset comes from simulator output we know that in this case noise or measurement error is not an issue, so the need for regularisation is limited. Similarly, whilst we have a large number of weights in our equation, the size of our training set is very large compared to this, which already acts to limit overfitting. Because of this we find only very small values are necessary.

## 2.3   Sensitivity studies

We wish to investigate the sensitivity of the regressor to its inputs, in order to understand the ways in which the regressor

is making its predictions. We do this in three ways. Firstly we directly assess the coefficients (weights) used in the resulting regressor. This indicates which features are being most heavily *used* in the predictions. Secondly, we run a series of withholding experiments, this indicates which inputs are most *necessary* for accurate forecasts. Lastly, for the inputs which the withholding experiments identified as being most critical to forecasts, we assess the impact these have on errors, giving insight into *how* these inputs effect the forecasts.

We assess the coefficients simply through plotting a heat map of coefficients (Fig. 4 and Sect. 4.1). Inputs that are highly weighted by the regressor (those with large coefficients) are important to the prediction, whereas those with low weights, can be considered as less important for the predictions.

Alongside this, we run a series of withholding experiments (Table 1 and Sect. 4.2). For each of the variables described in Sect. 2.2, with the exception of temperature, we train a new regressor leaving out that one variable group, e.g. we train a

new regressor with all the existing inputs except for salinity at all input points and any multiplicative terms including salinity. This corresponds to running the first pass of a Backward Sequential Search interpretability analysis. We also run two further withholding experiments. In the first we assess the importance of providing information in the vertical neighbourhood of points. Instead of the 3-d stencil originally used, we take a 2-d neighbourhood of points, 3x3, in only the horizontal direction, thus giving 9 inputs for each of temperature, salinity, etc. Lastly, we also run without multiplicative terms, i.e. the model consists





of only the first term in Eq. 1, giving a purely linear equation, enabling us to assess the impact of non-linearity on predictions. The new regressors are trained in exactly the same way, using the same training and validation samples - the only difference being the number of input features used. Comparing results from these withholding experiments to the control run show the importance of the withheld variable - if error increases significantly then the variable is necessary for accurate predictions. However if the impact on error is small, the regressor is able to make predictions of similar accuracy with or without that

variable, indicating it is not needed for good predictions.

Whilst these two methods (coefficient analysis and withholding experiments) both help to indicate the feature importance in the model, it should be noted they highlight different aspects of the importance of the input features. Looking at the coefficients of the trained regressor helps to identify which inputs are being most heavily *used* for the predictions from that particular regressor. By contrast, the withholding experiments indicate which variables are *necessary* to get predictions with the level of

accuracy shown in the control. There may, for example, be scenarios where certain variables are heavily weighted and flagged as important through the coefficient analysis, but when these same variables are withheld, the regressor re-weights other variables during the training step and maintains a similar level of accuracy due to correlations and the strong co-dependency of ocean dynamics on multiple variables. Coefficient analysis helps us to understand how a particular instance of a regressor is working, whereas the withholding experiments help us to understand the impact and importance of each variable in creating skilful

regression models more generally.

Lastly we analyse the resultant models from the three worst performing withholding experiments. We look at scatter plots of truth against prediction, and spatial plots of averaged absolute error to see how these models perform. We compare the average error plots to average errors in the control run (a run with all inputs) to see where errors are increased. We then compare this with the dominant processes driving temperature change in those regions (Fig. A1 and A2) and our expectations based on prior

knowledge of ocean dynamics to assess if the regressors respond in the ways we expect.

## 3   Regressor performance



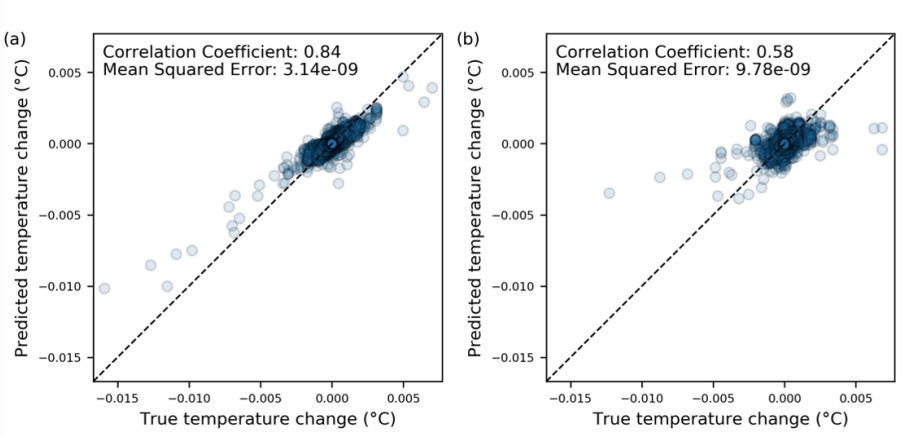

**Figure 2.** Scatter plot of predictions against truth for both training (left) and validation (right) datasets for the control regressor

Over the training set the regressor does a good job of predicting for both the dominant near-zero behaviour, and the very rare temperature changes of more than $\pm 0.002°$. Over the validation dataset the regressor drops in accuracy, with a tendency to under-predict, particularly for large changes, but still shows some skill.





First we discuss the performance of the control model — the regressor which is trained using the full set of previously discussed inputs. The predictions from the regression model closely resemble the true change in daily mean temperature in both the training and validation datasets (Fig. 2) although there is a tendency to under-predict the magnitude of temperature
changes. The model captures the central part of the distribution well. To a slightly lesser extent it also captures the tails of the distribution, where temperature changes are larger, although the under-prediction is more significant here. However, it is noteworthy that the model still shows considerable skill for these points, given that there are a relatively limited number of training samples in the tails — of the over nearly 650 000 training samples, just over 500 of those samples have temperature changes in excess of $\pm 0.001°$. Despite the relatively rare nature of these larger temperature changes, we feel that capturing these
alongside the smaller changes is critical to building a robust model. The underlying dynamics of the system, which we hope the regression model is able to learn, drives the full range of temperature changes seen. As such if we build a regressor which is unable to capture the extreme levels of change, this would indicate the model is not fully learning the physical dynamics as was intended. Capturing these extremes is also critical to obtaining a model which could (with further development) lead to a feasible alternative forecast system. While these extremes are limited in their occurrence, their impact on the ocean system
is notable, particularly if we were interested in longer prediction timescales. A model unable to capture these fails to provide a useful starting point for development of an ocean forecast system. Although we note that the development of a data-driven forecast system is not the focus of this work, the ability of the model developed here to capture extremes is to some extent relevant from that perspective.

Table 1 shows RMS errors and correlation coefficients for this run (top row) in comparison with a persistence forecast
(bottom row). A persistence forecasting is a forecast of no change - in this case to forecast zero temperature difference. Persistence forecasts are commonly used as a benchmark in forecasting, and already provide a statistically good predictor here due to the limited temperature change across most of the simulator domain. However, we can see the regressor performs significantly better than persistence. As expected we can see from Table 1 and Fig. 2 the regressor performs less well over the validation dataset, however it still outperforms the persistence forecast.



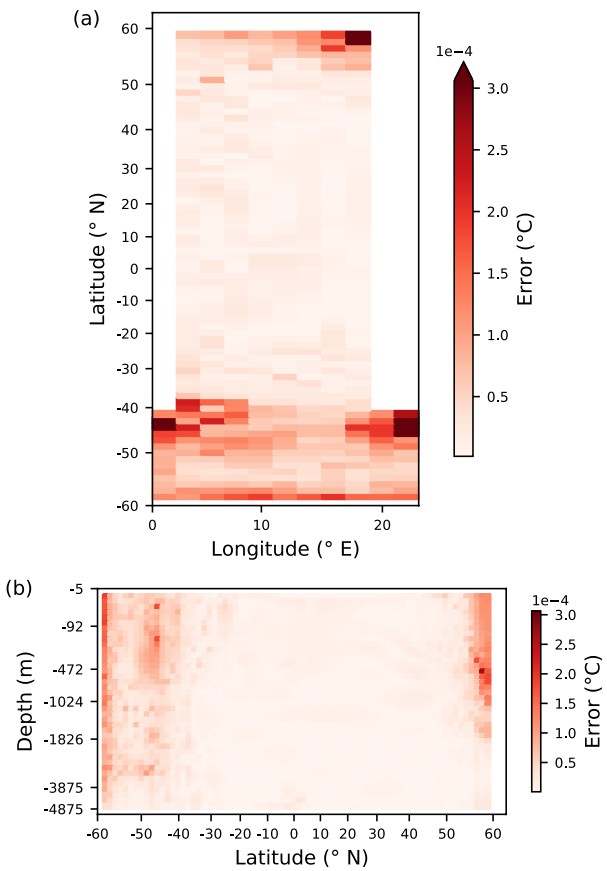

**Figure 3.** Mean Abs Error of predictions (°C) at -25m depth (top) and 13°East (bottom).

The errors are largest in the very north of the domain, and in the southern region, in locations where the temperature change itself is largest.

Comparing with Fig A1 and A2 we see errors are largest in areas of increased vertical fluxes and locations with high meridional diffusion, and high zonal advection.





## 3.1 Spatial patterns of errors

We calculate temporally averaged absolute errors to give us an indication of how the regression model performs spatially. These averages were created by taking the MITgcm state at 500 different times from the 20 year dataset and using these fields as inputs to the regressor to forecast a single time step ahead. The set of forecasts created from these 500 input states are compared to the truth from the GCM run, and the absolute errors between the truth and predictions are then temporally averaged. To emphasise, this is an average of 500 single time-step predictions, and not an average from an iterative run.

The set of input states spans the full 20 year MITgcm dataset, but with sub-sampling to take every 14th day (as opposed to every 200th day as was used in creating the training and validation sets). This results in a far larger set of input states than present in the training and validation data. The results here are therefore not specific to either the training or validation set, but instead show performance over a larger dataset which shares occasional samples with both.

These averaged errors are shown in Fig. 3. Note the regressor is only applied away from boundary and land points (in its current form it cannot deal with spatial locations which are not surrounded on all sides by ocean points) hence points close to land are not included in these plots.

Figure 3 shows the largest errors are located in the north of the domain and in the Southern Ocean. These are regions where the temperature change itself is largest (cf with Fig. 1 which show snapshots of daily temperature change), as would be expected. In particular, the large errors throughout the Southern Ocean section of the domain persist through depth, although the largest errors are associated with points above 1000m, or at the very southern extent of the domain.

Comparing Fig. 3b with Fig. A1 and A2 we see that the errors in the north of domain are co-located with regions of high vertical advective temperature fluxes, and regions of high convective fluxes. These results imply the regression model struggles to fully capture the vertical processes, and the associated heat flux, in the north of the domain. The high errors in the Southern Ocean are again co-located with regions of high vertical diffusive fluxes, this time both explicit and implicit, and vertical advection. Although, the pattern is less clear here, as the location of these errors is also a region of high meridional diffusive fluxes, and high zonal advective fluxes. Throughout the ocean interior where temperature changes and the fluxes associated with these are small, errors are also small, as would be expected.

The results are promising given the limitations of this model. Although we allow second order polynomial interactions, we are still working with a very simple regression model, and the order of complexity is no-where near that considered to be present in the simulator, or the physical ocean. To truly capture the dynamics of the ocean, far higher levels of interaction and complexity would be required. That a simple regressor achieves this level of skill is promising when considering the potential for applications of more complex data driven methods such as the neural networks described in Scher (2018), Dueben and Bauer (2018), Weyn et al. (2019), Arcomano et al. (2020), etc.

# 4 Sensitivity of the regressor

## 4.1 Coefficient analysis





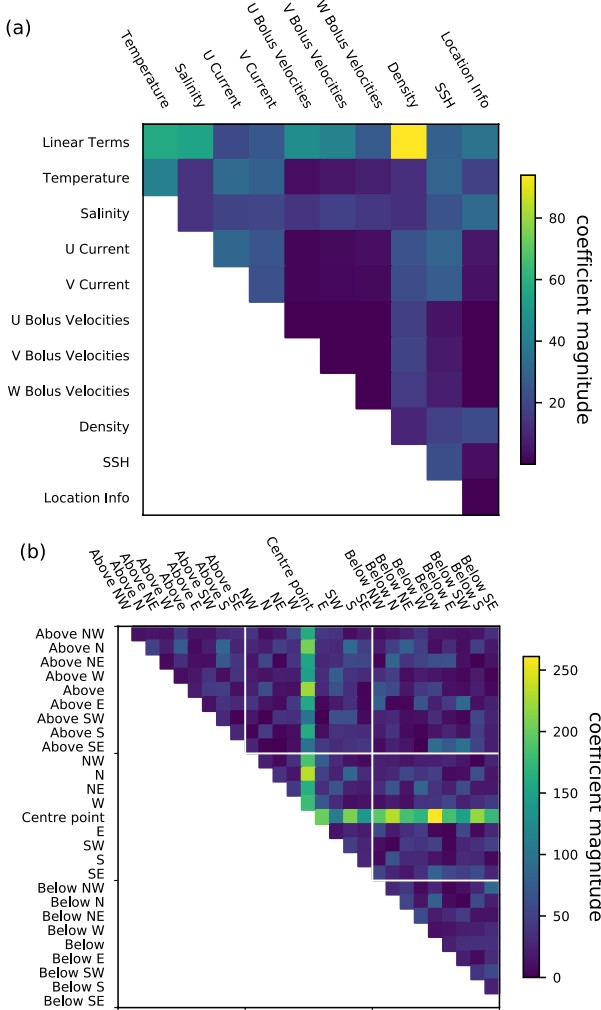

**Figure 4.** Coefficients of the control regressor. Top: Coefficients averaged over all input locations for each variable type, and each set of non-linear combinations of variables. Bottom: Coefficients for Temperature-Temperature interactions, with values for each input location and for each interaction between input locations shown.

We see that density is very heavily weighted, and therefore providing a large part of the predictive skill of this model, this is in line with our physical understanding that density changes are driving convective temperature change. The interactions between temperature at the point we are predicting, and temperature at surrounding points are also very highly weighted. This is line with our physical knowledge of advection and diffusion driving temperature change.





First we assess the sensitivity of the trained regressor by direct coefficient analysis. Figure 4 plots the magnitude of the coefficients in Eq. 1. Figure 4a shows coefficients averaged over all input locations for each variable type (i.e. for most variables there are 27 inputs, corresponding to the 27 neighbouring cells. We average over these to give a single value for each variable

(temperature, salinity, etc), and for each polynomial combination of variables). Figure 4b shows the coefficients related to polynomial interactions of temperature with temperature — these are the raw coefficients, without any averaging applied. High-weighted inputs (those with a large magnitude coefficient) are variables which are heavily used in the predictions and therefore considered important for the predictive skill, whereas inputs with low magnitude coefficients can be considered less important. Again, we emphasise that the coefficients highlight which features are being predominantly *used* in this model, but

this is not necessarily what is *needed* to create a skilful model — for that we need to look at the withholding experiments.

From Fig. 4a, we see that density (as a linear term, not in combination with other variables) is by far the most highly weighted variable in this model. The regressor is using density information as a very large component of its predictions. This is promising, as from our physical understanding of the system we know density is key to ocean dynamics. Unstable density profiles contribute to the large temperature changes seen in the south and very north of the domain, and for geostrophic currents

the flow follows the density stratification.

More generally we see that the location information is low weighted, particularly when interacting with other variables. This indicates the regressor is not basing its predictions predominantly on the location of points, but on the physical variables themselves.

From Fig. 4b we see that the multiplicative interaction between temperatures at different input locations are very highly

weighted for certain combinations of locations. Specifically, it is the interaction between temperature at the grid point we are predicting for and temperature at all surrounding points which gives the bright banding. This fits well with our physical expectation of the system — as diffusive and advective fluxes of temperature are dominated by local gradients in temperature.

## 4.2   Withholding experiments





**Table 1.** Table showing RMS errors and correlation coefficients for a series of withholding experiments. Results are also included from a persistence model (bottom row) for comparison. The two withholding experiments which make the largest difference to each error measurement are shown in red.

These are ordered in terms of RMS error over the training dataset, with variables which are most *necessary* for predictive skill appearing nearest the bottom. It is critical to include polynomial interactions. Information on the vertical structure, and on the currents is also necessary for good predictive skill.

|  | RMS error (°C) | | RMS normalised by control | | Correlation coefficients | |
| --- | --- | --- | --- | --- | --- | --- |
|  | Training | Validation | Training | Validation | Training | Validation |
| Control (full inputs set) | 5.61e-05 | 9.89e-05 | - | - | .84 | .58 |
| Withholding longitude | 5.65e-05 | 9.92e-05 | 1.01 | 1.00 | .83 | .58 |
| Withholding depth | 5.66e-05 | 9.91e-05 | 1.01 | 1.00 | .83 | .58 |
| Withholding latitude | 5.66e-05 | 9.94e-05 | 1.01 | 1.01 | .83 | .57 |
| Withholding salinity | 5.82e-05 | 1.01e-04 | 1.04 | 1.02 | .82 | .56 |
| Withholding density | 5.82e-05 | 1.02e-04 | 1.04 | 1.03 | .82 | .56 |
| Withholding SSH | 5.89e-05 | 1.01e-04 | 1.05 | 1.02 | .82 | .56 |
| Withholding bolus velocities | 7.32e-05 | 9.65e-05 | 1.30 | .98 | .70 | .55 |
| Withholding currents | 8.16e-05 | *1.07e-04* | 1.45 | *1.08* | .61 | *.39* |
| Using a 2-d (3x3) input stencil | *8.52e-05* | 1.06e-04 | *1.52* | 1.07 | *.55* | .40 |
| Without polynomial interactions | *1.02e-04* | *1.14e-04* | *1.82* | *1.15* | *.12* | *.11* |
| Persistence model (for comparison) | 1.02e-04 | 1.15e-04 | 1.82 | 1.16 | - | - |





RMS errors and correlation coefficients from a series of withholding experiments are shown in Table 1, along with results
from the control and a persistence forecast. Withholding experiments quantify the relative necessity of each input variable. The
larger the relative increase in error between the control and a withholding experiment, the more necessary that withheld feature
is for making accurate predictions. All withholding experiments perform at least as well as the persistence model (which is used
as a benchmark in weather and climate models) over the training and validation datasets, indicating that even with incomplete
input sets the regression models show significant skill. Correlation coefficients in particular show marked improvement over
the persistence forecast for all experiments over both training and validation datasets.

### 4.2.1   Withholding location information

The inputs which have the smallest impact on training error are those giving location information about the grid point being
predicted (the longitude, latitude, and depth of the grid cell). These variables have no direct influence on the dynamical pro-
cesses driving temperature changes in the simulator (note that whilst latitude has physical relevance in ocean dynamics due to
it being directly linked with the Coriolis effect, this does not directly drive temperature change — its impacts appear through
changes in velocities, which are provided to the regressor already). That the regressor performs well even when the model
has no location information indicates that well performing regressors are not heavily dependent on learning patterns which are
non-physically based on location, but may instead be learning patterns based on the underlying dynamics.

### 4.2.2   Withholding physical variables

The physical ocean variables have a higher impacts on error than the location variables — indicating that the regressor requires
knowledge of the physical system in order to make its predictions. Of these, withholding salinity, density or sea surface height
information has minimal impact. Again, these variables have limited direct influence on temperature — their effects are felt
through the resulting changes in currents caused by interactions of these variables. In a model able to capture more complexity,
or looking at forecasting over longer time periods, these variables may become more relevant, however when looking at
evolution of just temperature over a single day, they are of little direct importance, both physically and when developing skilful
regression models.

While density was heavily weighted as a coefficient, when withholding density the impact is small, especially when com-
pared to the impact of currents. This highlights the usefulness of interpreting models through a variety of techniques, each of
which give insight to different aspects of the way the model is working. The density of seawater depends on its temperature
and salinity, and so is tightly coupled to both of these. While the control model used density strongly in making its predictions,
when density is withheld the model has the ability to adjust by using these tightly coupled variables more heavily, enabling it
to still provide accurate predictions. This tight coupling and inter-dependency of density with other variables likely explains
the small impact seen in these experiments. The combination of information from the two methods used to analyse feature
importance indicates that density information is very highly used by the model when available, but that its usefulness can be
easily compensated by other variables if it is not provided to the model, i.e. it is sufficient but not necessary for model skill.





The experiment withholding the currents performs the worst of all the experiments concerning physical variables. That currents are one of the most important inputs required for regressor performance implies that some understanding of advection in the regression model is critical for accurate results, in line with our knowledge of the physical system being modelled. Errors from this experiment are analysed in more detail in Sect. 4.3.

### 4.2.3 Withholding vertical structure and multiplicative terms

The withholding experiments which have the highest impact on training error are those which train on only a 2-d stencil, or include only linear terms. Again, these experiments are analysed in further detail in Sect. 4.3.

Using a 2-d stencil means the regressor has no information about the ocean vertically above and below the location being predicted, and cannot use the vertical structure of the ocean in its prediction. We know this information to be important in the dynamics of the simulator, particularly in the south of the domain and the very north where vertical processes driven by the MOC affect temperature, and so it is reassuring that withholding it has such a large impact on error.

By restricting the regressor to purely linear terms (withholding polynomial interactions) we see the largest increase in error over the training set. That this purely linear version of the regressor performs poorly is also expected given our physical understanding of the problem being modelled. The ocean is known to be a complex, highly non-linear system, and we would not expect a purely linear regressor to be able to accurately replicate the detail and variability of these complex interactions.

### 4.2.4 Summary of withholding experiments

These withholding experiments emphasise the need for a regression model to have information on currents and vertical structure, as well as enough complexity to capture some of the non-linearity of the system in order to provide even a basic level of skill in forecasting temperature change in the ocean. The feature importance displayed here by the regressor is consistent with the importance of these inputs in the dynamics system we are modelling, implying the model is dependent upon the variables we would expect. Therefore, we are confident that the regressor is, to some extent, learning physical constraints rather than purely statistical patterns that might lack causality.

### 4.3 Further analysis of withholding experiments

We further investigate the results of the three worst performing models from the withholding experiments; withholding information on the currents, providing only 2-d inputs, and a purely linear model. We look closely at the model predictions and compare these with the control run to infer how the variables are impacting predictions.



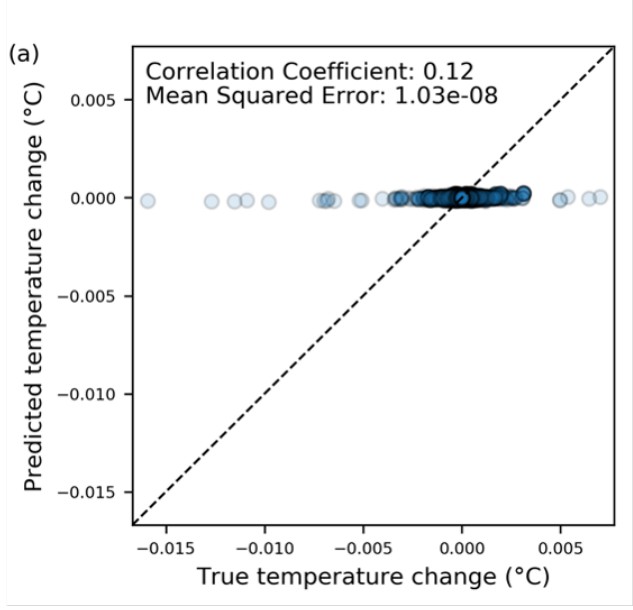

**Figure 5.** Scatter plot of predictions against truth over the training dataset for the regressor trained with no polynomial interaction terms

A purely linear regressor (trained without non-linear interactions) is unable to capture the behaviour of the system. This is expected as we know the underlying system to be highly non-linear.



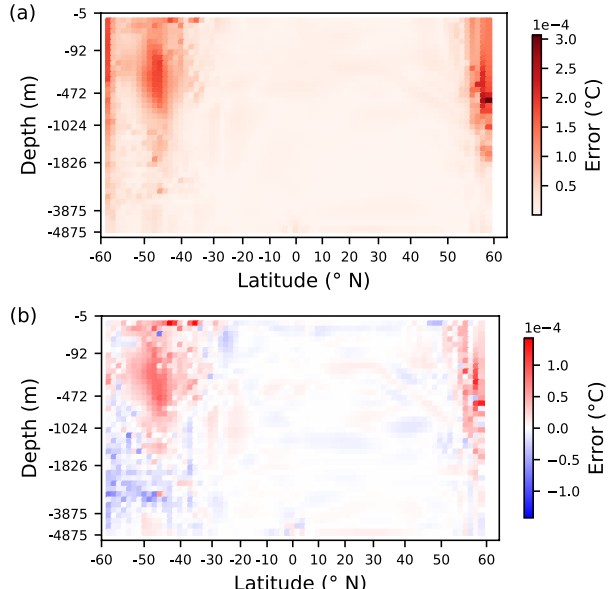

**Figure 6.** Top: Cross section at $13°$E of Mean Abs Error for the regressor trained using a 2-d stencil. Bottom: Difference between this and the control run (Fig. 3b)

When withholding information about the vertical structure, errors in the regressors prediction are increased in a region north of $50°$ and south of $-30°$. Comparing this with Fig. A1 and A2 we can see how the areas of increased errors correspond to particular processes.

### 4.3.1 Impact of multiplicative terms

Figure 5 shows the performance of the purely linear model, that is the model trained without any multiplicative terms. We see that without multiplicative terms the model can capture the mean behaviour of the system (zero change in temperature) but is
unable to capture any of the variability. This mean behaviour alone does not provide useful forecasts, as can be seen from the statistics, particularly the correlation coefficient for this experiment. Comparing Fig. 5 with Fig. 2 we see the importance of the non-linear terms in predicting temperature change, especially for samples where temperature change is non-zero. Non-linearity is shown to be critical to modelling the variability of temperature change.





### 4.3.2 Impact of vertical structure

To assess how information about the vertical structure of the ocean impacts predictions we look at spatially averaged errors from the model trained with only a 2-d halo of inputs, along with the difference in error between this and the control run (Fig. 6). This top plot is created in the same way as Fig. 3b, with the absolute error from predictions across the grid at 500 different times averaged to give a spatial pattern of errors. The bottom plot shows the difference between the top plot and Fig. 3b, with areas shaded in red indicating where the error has increased as a consequence of withholding information about the vertical
structure, and blue indicating areas where the predictions are improved. By comparing Fig. 6b with Fig. A1 and A2, we can see which processes are dominant in the regions of increased error, and make inferences about the ways in which the additional inputs are being used in predictions.

   Interestingly this regressor shows some regions (the deep water in the south of the domain) where the errors are notably improved in a regressor using only 2-d information. Given the regressor we have developed is learning one equation to be
applied across all grid boxes in the domain it is not unexpected that removing variables might improve performance in a few small regions if the restricted equation is a better fit for that particular location, but not more favourable across the entire domain. This highlights the limitations of our method, and the need for more complex data-driven models which can better adjust to the wide variety of dynamics shown across the domain.

   More interestingly, we see that using a 2-d stencil rather than a 3-d stencil increases errors in the very north of the domain,
and in a region south of -40°. The area of increased error in the north coincides most closely with a region of high convective fluxes. We note that it also corresponds to a lesser extent with a region of high vertical advection, however the shape and location near the surface seem to far better correspond with the region of high convection. Convective activity is driven by dense water overlying less dense water leading to vertical mixing. For the regressor to 'learn' the change in temperature associated with this, it would require information about the vertical density profile. That errors are increased in this region when information
about the vertical structure is withheld, implies that the model is dependent on the vertical structure in the ways we would expect.

   The increased errors seen in the upper waters of the Southern Ocean are more complicated. They are roughly co-located with regions of high zonal advection and high meridional diffusivity. This is unexpected, given these are horizontal processes and should not depend on the regressor having knowledge of what is happening above and below the point being predicted. How-
ever, we can see from Fig. A1 and Fig. A2 that the Southern Ocean is a region of very complex dynamics (considerably more so than other regions in this GCM configuration), with many different processes occurring. Within this complex dynamical region, there are clear signals of high vertical diffusive fluxes and convection, which would be more in line with our physical expectations, although these appear far broader than the specific regions of increased error which we see. It may be that the increase in errors in this region is driven by the regressors reduced ability to capture the vertical diffusion and convection, as
would be in line with our physical expectations. However, these results more strongly indicate that the regressor is learning spurious links between the inputs provided for a vertical neighbourhood of points, and zonal advection and meridional diffusion. It should be emphasised that the complex dynamics of the Southern Ocean may test the limitations of such a simple





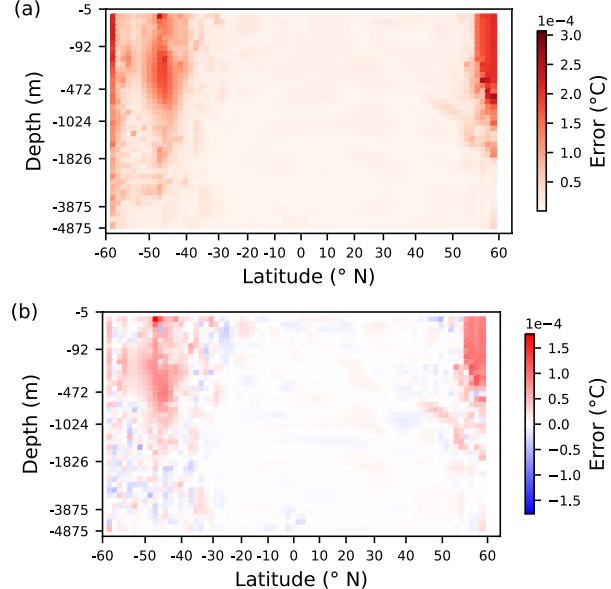

**Figure 7.** Top: Cross section at 13° E of Mean Abs Error for the regressor trained with information on the currents withheld. Bottom: Difference between this and the control run (Fig. 3b)

When withholding currents, errors in the regression model are increased north of 55°, and in a broader region south of -35°. Comparing this with Fig. A1 and A2 we can see how the areas of increased errors correspond to particular processes.

regressor, causing the model to revert to less-physically relevant patterns in this area. In particular, in this region currents flow along non-horizontal isoneutral surfaces, meaning there is inherent interaction between the processes considered here. It may
well be the case that such a simple model is not able to capture this interaction, and a similar assessment performed on a more complex data-driven model would be of interest here.

It is important to emphasise that this analysis only infers plausible explanations, it is not able to definitively attribute the increased errors to any specific process. We see here that there are very plausible explanations for the errors seen in the north of the domain, which are in line with what we expect from a regressor which has learned the underlying dynamics of the ocean.
By contrast, while there are physically consistent explanations available for the increased errors in the south of the domain, there are stronger indications of less physically consistent behaviour. This implies that in the complex Southern Ocean region, the regressor struggles to fully capture the dynamics of the region, particularly with regards to the impact of using information on the vertical structure of the ocean.





### 4.3.3 Impact of currents

We analyse the impact of the currents on the regressor by again looking at the locations where errors are increased between this experiment and the control run, and comparing these to the dominant processes in those areas. Figure 7 shows the spatially averaged errors from this regressor along with the difference between these and the errors from the control model.

The horizontal (U and V) components of the currents directly drive horizontal advection of temperature. They are also indirectly related to horizontal diffusion, as this is increased in regions of high currents and steep gradients. As such, we would
expect that suppressing information about the horizontal currents would cause increases in error in regions where horizontal advection and horizontal diffusion is high. Comparing Fig. 7b to Fig. A1 and Fig. A2, we do indeed see a region of increased error south of -40°, which coincides with the regions of high zonal advection and high meridional diffusivity. However, again we note that this region of increased error is one where many processes are present, and the increased errors seen also coincide, to a lesser extent, with regions of high vertical processes (advection, diffusion and convection), which is not in line with our
physical understanding. Here, errors appear more closely matched to the horizontal processes, and so a reasonable interpretation is that the model here is behaving as expected, though again we emphasise that it is not possible, based on the evidence here, to definitively attribute the increased errors to any specific process, only to make plausible inferences.

The largest increase in errors are in the very north of the domain — an area where the temperature flux is dominated by vertical processes, both vertical advection (driven by vertical currents) and convective activity (i.e. due to instabilities in the
water column). The increased errors in this northern region seen in Fig. 7b seem to most closely correspond with the region of large vertical advection seem in Fig. A1c. Whilst it may at first be unintuitive that errors are increased in a region dominated by *vertical* advection when *horizontal* currents are withheld, this is in fact in line with our understanding of the dynamics of the system. Vertical advection is indirectly linked to the horizontal currents, as vertical currents are predominantly a consequence of convergence or divergence of the horizontal flow (particularly as the vertical flux of water driven by vertical motion as a
result of unstable density profiles manifests in the convective fluxes). The results here imply that as the regressor is not directly given information on the vertical currents, it may be learning the link between the horizontal and vertical currents, and the resultant vertical advection. Without information on the horizontal currents, the regressor struggles to capture this vertical advection resulting in increased errors in this northern region, in line with our understanding of the physical processes being modelled. It is noteworthy that the increase in errors here are larger than those in Fig. 6. However, if our hypothesis is correct,
the errors here are associated with vertical advection and the errors in Fig. 6 are associated with convection, than the different contributions to heat flux of these two processes (see the scales on Fig. A1 and A2) explains the smaller change in errors seen here.

### 5 Conclusions

We have shown a regression model, despite being a simple statistical tool, is able to model change in daily mean temperature
from an ocean simulator with notable skill when appropriate inputs are provided. That such a simple data-driven method can make skilful predictions gives promise to the growing set of data-driven approaches for weather and climate modelling. One





concern around these methods is the lack of physical basis and that might limit the ability for these models to perform well 'out of sample' (that is over datasets outside of the training region). For the regression model we have shown that the sensitivity of the model outputs to the model inputs is generally in line with our physical understanding of the system.

Specifically, we analyse the coefficients of our regression model and find that the predictions for a grid cell are based heavily on the density at the surrounding points, and the interaction between temperature at the grid cell and its neighbouring points. The importance of temperature interaction at surrounding points is representative of advective and diffusive processes which take place across the domain. The importance of density is in line with the the simulator representing, to some extent, density driven currents which are responsible for much of the changes in temperature in this GCM configuration. While later

withholding experiments show that density is not *necessary* for skilful predictions, this is most likely due to the dependency of density on temperature and salinity, and the regressors ability to use these variables in place of directly using density. Again, this behaviour makes sense when considering the physics of the ocean.

    We conduct a number of withholding experiments. These show that withholding information about the location of the grid cell being forecast has very little impact on accuracy. In contrast, withholding information on the physical ocean variables has

a larger impact. Of these, the velocities have the biggest impact, in line with our knowledge of advection being a key process in the transfer of heat. We see that both inclusion of non-linear interactions between inputs, and information about the vertical structure (rather than solely the horizontal structure), are needed for skilful predictions. Again this is inline with our knowledge of the physical system. The ocean is highly non-linear and it would be expected that a non-linear model is needed to capture its behaviour. Similarly, the ocean dynamics are inherently three dimensional, and so it is expected that inputs from a 3-d

neighbourhood are necessary for predictive skill.

    Further analysis of the three worst performing withholding experiments give insight into how these inputs impact predictions. We see that including some level of non-linearity is critical to capturing the complex nature of the system. Looking spatially at the errors from experiments that withhold currents, and withhold information about the vertical structure, we see that errors are generally increased in the locations that we would expect, and in ways which are in line with the known dynamics of the

system. The caveat to this is within the complex dynamics of the Southern Ocean. Here, although physically consistent results can be inferred, the patterns seen are complex, making it difficult to reasonably infer one particular scenario over another. It is not possible to definitively attribute increased errors to specific processes through this analysis, only to highlight plausible explanations, and in this this complex region multiple explanations can be inferred. This is especially notable when looking at the impacts of vertical structure in the Southern Ocean region. Here the evidence more strongly indicates physically inconsistent

inferences, indicating the regressor has struggled to learn the full dynamics of this region. Nevertheless, it is reassuring that in most cases, and especially when looking at the north of the domain where the dynamics are less complex, physically consistent interpretations can be seen.

    Our results highlight the need to perform model interpretation through a variety of methods, assessing both feature importance within models, and the ways in which features impact predictions. Generally we see that the regressor developed here

both utilises and depends on variables which are in line with the known dynamics of the system, and these variables impact predictions in ways which are consistent with our physical knowledge. These results imply that the regressor is, to a large





extent, learning the underlying dynamics of the system being modelled. This result is very promising in the context of further development of data-driven models for weather and climate prediction, for both atmospheric and oceanic systems. That we see this behaviour in a simple model suggests that more complex models, capable of capturing the full higher-order non-linearity

inherent in GCMs, are well placed to learn the underlying dynamics of these systems. As data-driven models become competitive alternatives to physics driven GCMs, it is imperative to continue to investigate the sensitivity of these models, ensuring we have a good understanding of how these models are working and when it is valid to rely on them.

*Code and data availability.* Code used for this work (analysing the MITgcm dataset, and training and analysing the regression models) can be found at https://github.com/RachelFurner/RegressionModelOceanTemp/tree/v1.0 The MITgcm dataset used is available at https:

//zenodo.org/record/4672260 (DOI 10.5281/zenodo.4672260)

**Appendix A:  GCM Fluxes**



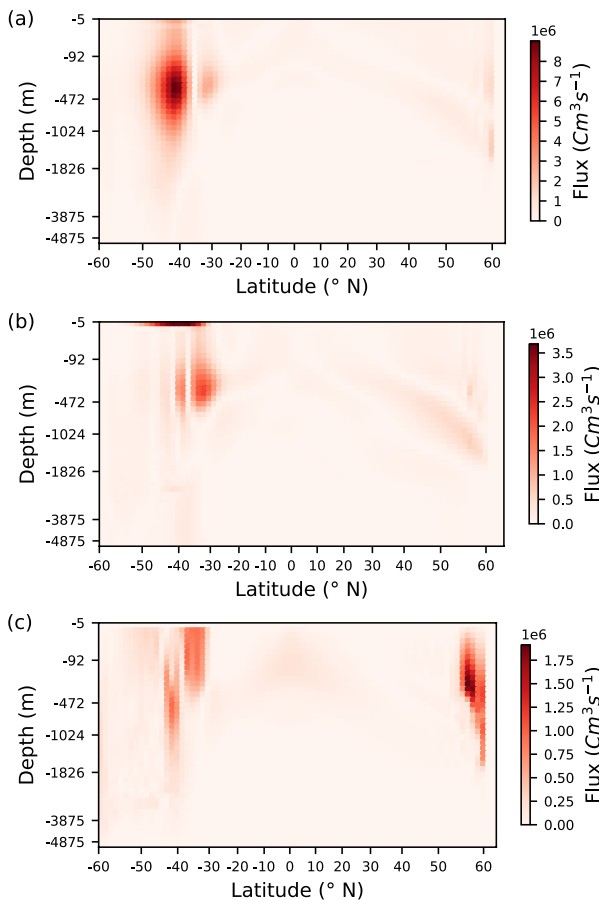

**Figure A1.** Average absolute zonal (a), meridional (b), and vertical (c) advective fluxes of temperature at 13° E.

Horizontal advective fluxes are largest in the southern region of the domain, associated with the ACC-like current. There's a large amount of vertical advection in the north of 55°, and at -30 to -40°, associated with regions of upwelling and downwelling.



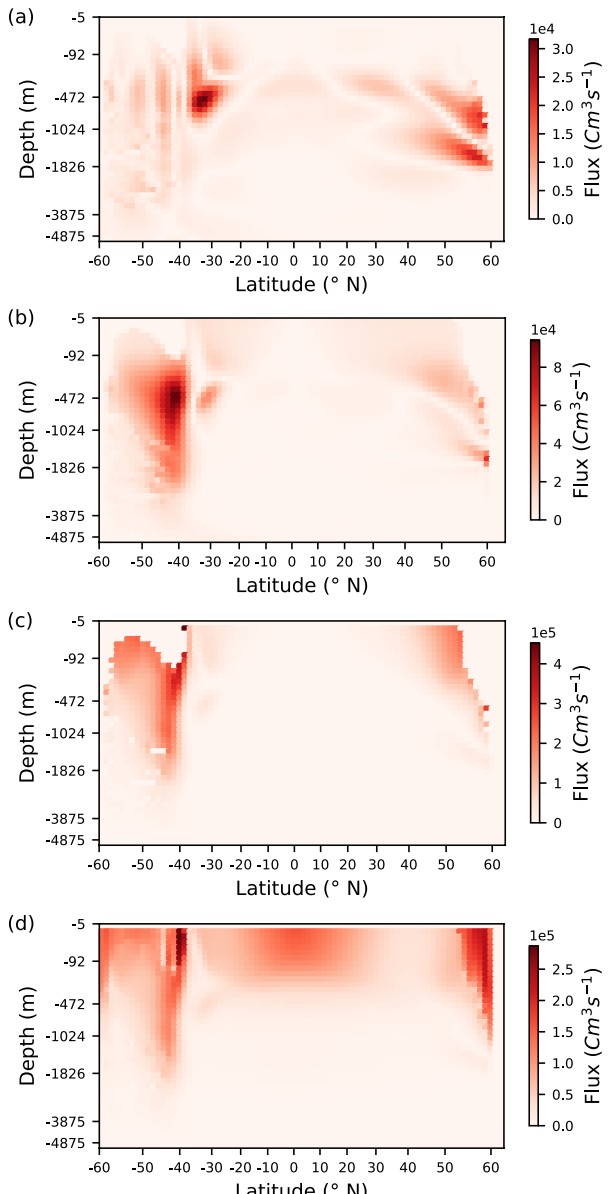

**Figure A2.** Average absolute zonal (a), meridional (b), and (explicit) vertical (c) diffusive temperature fluxes, and convective (implicit vertical diffusive) temperature fluxes (d) at 13° E.

There are large amounts of meridional diffusion associated with the ACC-like jet in the south. Zonal diffusion occurs in mid depth in the north of the domain, and just north of -40°. Vertical diffusion occurs through the south of the domain, and a small region just south of 50°. Convection occurs throughout the domain, and is particularly noteworthy in the upper waters of the ocean north of 50°, and south of -35°.





We calculated temporally averaged advective and diffusive transports of temperature to identify which processes dominate temperature change in different regions of the domain. Figures A1 and A2 show cross sections of these transports. These are created using the same data as used in Fig. 3, 6 and 7. They show an average of 500 daily transport tendencies, taken from the
20 year model dataset described previously, subsampled to take the average of every 14th day.

From these we see that the majority of temperature change from all processes is located in the Southern Ocean and the very north of the domain. In particular, we see the vertical advection is largest in the very north, and increased at the edge of the Southern Ocean. There is notable zonal advection of temperature around $40°$S, in keeping with the high wind stress and interaction with the end of the land feature giving rise to a Southern Ocean jet — an Antarctic Circumpolar Current (ACC) like
feature. Diffusive fluxes are generally lower (by one or two orders of magnitude). These show a broader spatial spread, although vertical, zonal and meridional diffusive fluxes are still highest in the Southern Ocean and near the north of the domain. There is a large signal of convectively driven temperature change, due to the surface cooling in this area (applied through surface restoring). Similarly we see increased vertical diffusive fluxes, both implicit and explicit, in the south of the domain, this is a region of high vertical activity, with both upwelling and downwelling of water masses. In the south we also see a signal of
strong meridional diffusion related to the ACC-like feature in the GCM.

**Appendix B: Full set of Coefficients**



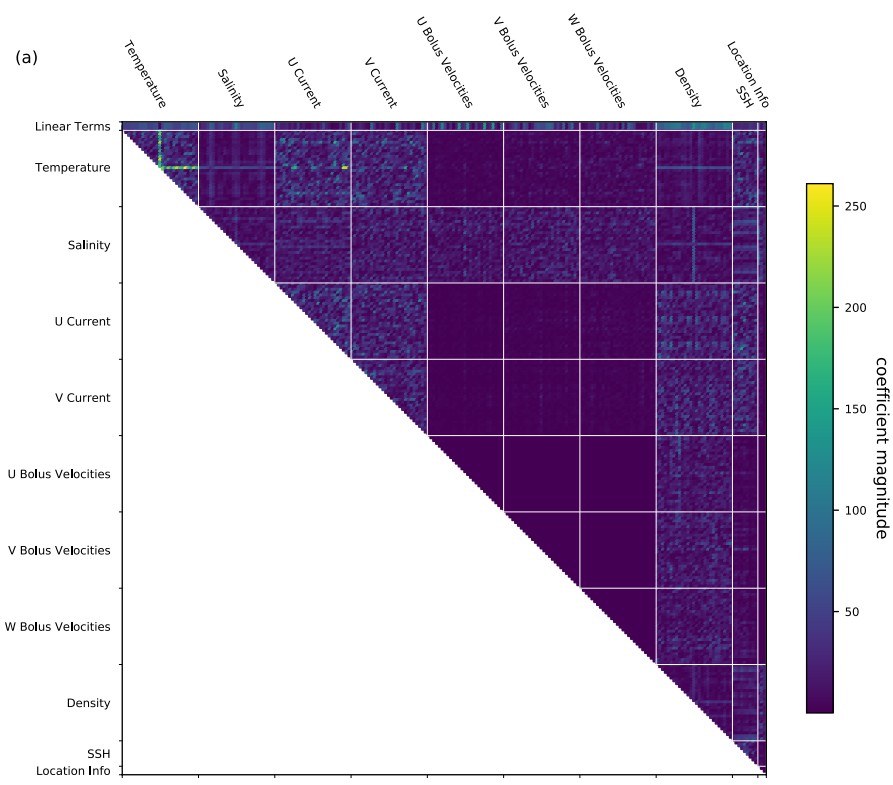

**Figure B1.** Coefficients of the control regressor for each input location and for each variable type. For linear inputs (top row) and for each set of non-linear combinations of variables.





Section 4.1 and Fig. 4a focused on coefficients averaged over each variable. Here, for completeness, we show the coefficients for all 26,106 inputs. The top row shows coefficients for the linear features (the first term in Eq. 1), and the triangular lower section shows coefficients for the non-linear terms (the multiplicative terms from the second term in Eq. 1). Note for most variables there are 27 pixels, once for each point in the 3-d neighbourhood stencil.

As in Fig. 4 we see the importance of density (as a linear term), the relative unimportance of location information, and the notable pattern in the multiplicative interaction between temperatures at different relative locations. Looking here at the full set of coefficients without any averaging applied we also note that the interaction of the U component of the current with temperature shows one or two points which are highly weighted. We suspect this is symptomatic of regression models trained by least squared error methods being highly sensitive to a few extreme training samples. When developing the regressor, we saw that different versions of the model all had a tendency to weight one or two coefficients very highly, with each iteration of the model favouring different coefficients. The more general patterns, particularly those seen when averaging over variables, and the sharp stripes in the temperature-temperature interactions, were consistent through all versions of the regressor, just the occasional very small number of highly weighted inputs changed. This indicates a lack of robustness in the model, meaning small changes to the training dataset can erroneously cause a few terms in the equation to become very highly weighted. This highlights the need to consider the robustness of data-driven models, and their sensitivity to the training samples used, and also emphasises the importance of using more than one technique to assess feature importance.

*Author contributions.* Plans for the fundamental design of the work and analysis methods were developed through conversations between all authors. The paper was predominantly written by RF, with contributions and comments from all authors. All programming was done by RF with guidance and assistance from other authors.

*Competing interests.* The authors declare that they have no conflicts of interest.

*Acknowledgements.* RF was supported by the UK Natural Environment Research Council [grant number NE/L002507/1]. DRM was supported by the UK Natural Environment Research council [ORCHESTRA, grant number NE/N018095/1]. DJ is supported by a UKRI Future Leaders Fellowship [grant number MR/T020822/1].





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
