# Peer review of "Sensitivity analysis of a data-driven model of ocean temperature"

_Geoscientific Model Development, 2021_

## Referee Comment (RC1)

In this paper, the authors present a polynomial regression model that data drivenly predicts the tendency of ocean temperature using the MITgcm ocean model. The study is well conducted, and technically correct. A lot of well-thought-out experiments have been conducted and presented reasonably well with enough details to facilitate reproducibility. The authors have even considered nuances such as correlated training-testing datasets and did a very comprehensive analysis. Having said all that, it must also be pointed out that the statistical model considered here is overly simple (and shows very limited skill of 0.58) and does not contribute to building better (not interpretable) data-driven forecasting models. The authors do acknowledge that throughout the length of the study and have even put an entire section talking about the limitations. In my opinion, although the study is very comprehensive, the model is too simple, and (as pointed out by the authors themselves) do not actually have the capability to iteratively (autoregressively) perform ocean temperature prediction as has been done (although in the context of atmosphere) in several papers that the authors themselves have cited, nor does it show good skill for even one-time-step prediction. Underneath, I list my major concerns with the current version of this paper and while these concerns do not question any technical point in the paper, the method presented does not quite serve as a "model" for geoscientific predictions and thus may not be suitable for publication in this journal.

1. The authors very thoroughly investigate the effect of different input variables in the one-time-step prediction of the temperature tendency, however, the bulk of the temperature difference between time step $t$ and $t + \Delta t$ is 0. Even for a simple polynomial model, this is probably a simple problem. While the authors claim that there is some skill in predicting high value of $\Delta T$, the plot also shows several such predictions missed by the model. This is not surprising owing to the simplicity of the model. *A correlation coefficient of* 0.58 *in validation* really shows that the model is just not skillful. Doing any analysis on such a model, in my opinion, may not be the best path forward in this field of machine learning/data-driven forecasting of weather/climate. Predicting extremes in the field of weather and climate has been done with data-driven models and authors should probably take a look at those literature, e.g., https://arxiv.org/abs/2103.09743, and https://agupubs.onlinelibrary.wiley.com/doi/full/10.1029/2019MS001958. The major issue that the authors face in predicting the extremes is essentially addressing the class-imbalance problem which is well studied in the ML literature.

2. The authors quite correctly point out that their model cannot do autoregressive/ iterative prediction thus rendering their model not-useful to some extent. Still, interpretability of these models is a big step forward in this field. However, doing so from looking at predictions at a single time step may not be the right approach. There have been several studies that have shown that error propagation in this model is non-trivial and nonlinear. Thus, data-driven models that iteratively forecast the state of the atmosphere/ocean may show variable skill based on how far it directly forecasts and the error analysis may lead to starkly different results e.g., Figure 2 of this paper (https://agupubs.onlinelibrary.wiley.com/doi/full/10.1029/2020MS002203) shows the effect of changing the time step of prediction in the data-driven model and so does this paper (https://gmd.copernicus.org/preprints/gmd-2021-71/) . While it is definitely true that a physical variable that actually affects ocean temperature would probably lead to better skill if used as an input, this observation has been reported in the conext of

atmospheric dynamics in a more complex deep learning model (https://agupubs.onlinelibrary.wiley.com/doi/10.1029/2019MS00705) and is quite intuitive to be honest. It is still worth mentioning that the authors do a remarkable job at presenting these analyses which are quite comprehensive in the context of this problem.

I would like to emphasize that the analyses conducted in the study are very comprehensive and would had been much more impactful had it been done for a model that was useful for data-driven prediction. Ofcourse, with an increase in complexity of the model, these analyses would become non-trivial as well. However, at this current stage, the best validation accuracy of the data-driven model is 0.58 which questions the performance of the model especially for a single time step prediction. One of the reasons could be because of under-predicting the extremes which can be dealt in other ways as well.

---

## Author Comment (AC3)

Many thanks for taking the time to read and review our paper, and for the positive comments on the technical aspects of the work, along with the helpful remarks and suggestions. We have responded to the comments in blue below (with the reviewers original comments in grey). Where we have made changes to the manuscript we've noted these below our comments, with red strikethrough text for deletions, green text for additions and black text indicating regions of no change.

In this paper, the authors present a polynomial regression model that data drivenly predicts the tendency of ocean temperature using the MITgcm ocean model. The study is well conducted, and technically correct. A lot of well-thought-out experiments have been conducted and presented reasonably well with enough details to facilitate reproducibility. The authors have even considered nuances such as correlated training-testing datasets and did a very comprehensive analysis. Having said all that, it must also be pointed out that the statistical model considered here is overly simple (and shows very limited skill of 0.58) and does not contribute to building better (not interpretable) data-driven forecasting models. The authors do acknowledge that throughout the length of the study and have even put an entire section talking about the limitations. In my opinion, although the study is very comprehensive, the model is too simple, and (as pointed out by the authors themselves) do not actually have the capability to iteratively (autoregressively) perform ocean temperature prediction as has been done (although in the context of atmosphere) in several papers that the authors themselves have cited, nor does it show good skill for even one-time-step prediction. Underneath, I list my major concerns with the current version of this paper and while these concerns do not question any technical point in the paper, the method presented does not quite serve as a "model" for geoscientific predictions and thus may not be suitable for publication in this journal.

We acknowledge the limitations you highlight, although we think that the work is still suitable for publication in this journal. We address the particular comment concerning the value of 0.58 as a measure of skill in a later reply below. Here we focus on more general principles.

We think that this work is useful to the scientific community. The model, while simple, is still a suitable framework for carrying out sensitivity analysis, which is a critical tool in the Earth system sciences. Added to this, our efforts provide a new use for data-driven methodology in ocean modelling. As far as we are aware, there are no examples of use of data-driven techniques being used in this way for ocean models; existing work focuses on atmospheric models or on using machine learning to improve components of ocean models. While the underlying physics is similar across atmospheric and oceanic systems, the applications and regimes of interest are often very different. Data-driven methods are best tested and analysed in both systems, building trust within both communities. We note the concerns over the predictive skill of the model and address these below, but even taking this into account, we think these results are interesting in themselves, and they provide a new baseline for further work, by both the authors and others, and a framework in which to begin addressing oceanographic applications.

Importantly, we note that while the model presented is not designed to be used in a forecasting sense, the GMD guidelines note that the journal considers manuscript types including 'geoscientific model descriptions, from statistical models to box models to GCMs'[1], indicating that even very simple models such as ours, which increase understanding rather than providing tools usable for real-world predictions, fit well within the scope of the journal. Added to this, GMD
* * *
[1] https://www.geoscientific-model-development.net/

scope also includes 'New methods for assessment of models'. Here we present an important aspect of assessment of data-driven models, which is again well suited to publication in this journal. While acknowledging the limitations and simplicity of this model, we think the work presented nonetheless provides a key step towards future capability, providing a 'proof of concept' for an oceanographic application and a baseline for further work. It also increases understanding and confidence in the growing field of statistical geophysical models, and therefore is of value to this journal.

We've made the following changes to the manuscript to help better emphasise the focus and relevance of the paper:

~Line 6:

> We develop a simple regression model of ocean temperature evolution, Ocean Temperature Regressor v1.0, and investigate its sensitivity to improve understanding of whether data-driven models are capable of learning the complex underlying dynamics of the systems being modelled, or if they instead learn statistically valid, but not physically based patterns.

~Line 78:

> ... the skill shown in using data-driven methods for predictions of the atmosphere suggest that these same methods could provide skilful predictions for the evolution of the ocean.
>
> The model developed here is highly simplified, both in terms of the idealised GCM configuration we train the model on, and the data-driven methods used. However, the underlying configuration captures key oceanic dynamics, enabling a suitable test bed to see if data-driven methods can capture the dynamical basis of these systems. Similarly, while we use a simple regression technique, this has sufficient skill to assess the ways in which the model works and improve understanding for the potential of data-driven methods more generally.
>
> We apply model interpretation techniques to our data-driven model to try to understand what the model is 'learning' and how the predictions are being made...

1. The authors very thoroughly investigate the effect of different input variables in the one time-step prediction of the temperature tendency, however, the bulk of the temperature difference between time step $t$ and $t + \Delta t$ is 0. Even for a simple polynomial model, this is probably a simple problem. While the authors claim that there is some skill in predicting high value of $\Delta T$, the plot also shows several such predictions missed by the model. This is not surprising owing to the simplicity of the model.

While much of the difference is near zero, very few points are actually zero, with the majority instead representing small changes to temperature. These changes, whilst small, are still very important to capture as they accumulate to give the large-scale motion seen in the ocean. Predicting these small changes is a key first step in being able to model the ocean.

A correlation coefficient of 0.58 in validation really shows that the model is just not skillful. Doing any analysis on such a model, in my opinion, may not be the best path forward in this field of machine learning/data-driven forecasting of weather/climate.

Whilst we are certainly not claiming large skill for the particular model being considered, we do emphasise the limitations of the correlation coefficient as a measure of skill and the resulting interpretation of the value of 0.58 as implying that there is no skill.

We note firstly that different statistical measures of skill provide different insights into the performance of the model. While correlation coefficients are useful, they are heavily influenced by extremes. Our main focus is on providing a useful estimate of increments in temperature taken across all model grid points, rather than capturing extreme values of increments that occur infrequently and at a small number of locations (see below for further comments on 'extreme values'). For this we feel that RMS statistics offer a more meaningful measure of skill. RMS errors are also a far more commonly used statistic when validating forecast models, and while the model developed here is not intended as a forecast model, our aim is similar in trying to represent the general behaviour of the ocean, and so here RMS errors are a very important measure of skill. We also note that it is important to interpret statistics in relation to some baseline. In short term prediction, across atmospheric and oceanographic applications, a common first baseline is persistence (Mittermaier 2008[2]). When comparing across RMS statistics, we see notable skill in the control when compared to a persistence forecast (an RMS of 9.89e-05 compared with 1.15e-04 over the validation set).

Regarding correlation coefficients, a value of 0.58 still implies that there is a substantial amount of useful information in the predictions -- especially when comparing that against the value zero which would correspond to a model with no useful information. Here it would still be best to compare to persistence however it is not trivial to obtain a correlation coefficient for a constant dataset (i.e. the persistence forecast data set, where $\Delta t = 0$ for every sample). We can however see from Fig. 5 that in the experiment where nonlinear terms are withheld predictions are comparable to a persistence forecast. Here the correlation coefficient is 0.11 over the validation set. Based on this, we can confidently infer that a correlation coefficient for a persistence forecast would be at most 0.11 (potentially even lower) over the validation set, and so again when comparing to this baseline the score of 0.58 for the control model shows considerable skill.

Importantly, we note that unlike similar papers in this area, we have trained the model to learn the temperature increment and assessed the model on these predicted increments, rather than the future temperature. While this should have minimal impact on RMS errors, correlation coefficients are hugely impacted by the framing of this question. Calculating correlation coefficients on these increments gives considerably lower scores than if we were to calculate them on the model's predicted field. Looking at both correlation coefficients and anomaly correlation coefficients calculated the predicted future temperature rather than the increment, these are very close to one. We have added the following to the manuscript to highlight this.

~ Line 279:

> As expected we can see from Table 2 and Fig. 2 the regressor performs less well over the validation dataset, however it still outperforms the persistence forecast. It should be noted that the regressor developed here is trained to predict the increment in temperature ($\delta T$), rather than the future temperature (T), and importantly is assessed on this increment prediction. If we assess predictions of future temperature, rather than predictions of the temperature increment, we see correlation coefficients and anomaly correlation coefficients very close to one (differing at the 9th and 6th decimal place respectively) over both the training and validation datasets.

Predicting extremes in the field of weather and climate has been done with data-driven models and authors should probably take a look at those literature, e.g., https://arxiv.org/abs/2103.09743, and
* * *
[2] Mittermaier, M. P. (2008). The Potential Impact of Using Persistence as a Reference Forecast on Perceived Forecast Skill, Weather and Forecasting, 23(5), 1022-1031.

https://agupubs.onlinelibrary.wiley.com/doi/full/10.1029/2019MS001958. The major issue that the authors face in predicting the extremes is essentially addressing the class imbalance problem which is well studied in the ML literature.

While the prediction of extremes is an interesting problem, with much relevant literature available, this is not the focus of our work. The intention here is to explore the potential for data-driven methods to eventually be used alongside (or instead of) traditional forecast models. Traditional forecast models are required to predict all behaviour, and predicting the more commonplace dynamics is often the first goal. Whilst our work is highly idealised, it is motivated by this perspective, and as such, here we instead aim to first predict the more common dynamics. Predicting extremes in the sense of the papers referenced here has a very different significance to our problem – not least because our extremes (unusually large values of increments) may have a limited effect on the overall evolution of the system. This would, however, make for interesting further work, and the authors comments highlight the need for data-driven forecast systems to capture these extreme dynamics alongside the more commonplace. We have made the following changes to the manuscript to better clarify these points.

~ Line 258:

The predictions from the regression model closely resemble the true change in daily mean temperature in both the training and validation datasets (Fig. 2) although there is a tendency to under-predict the magnitude of temperature changes. The model captures the central part of the distribution well. Whilst the majority of the temperature change is dominated by small near-zero changes, capturing these is key to producing a good forecast system. Although the complete development of a data-driven forecast system is not the focus of this work, we are motivated by the potential for data-driven methods to replicate traditional forecast systems. As such, the ability of the model developed here to capture the full range of dynamic behaviour, beginning with the most common dynamics, is key.

To a  lesser extent  the regressor also captures the tails of the distribution, where temperature changes are larger, although the under-prediction is more significant here. However, it is noteworthy that the model still shows  some skill for these points, given that there are a relatively limited number of training samples in the tails — of the over nearly 650 000 training samples, just over 500 of those samples have temperature changes in excess of ±0.001∘C, and the model used is very simple. Despite the relatively rare nature of these larger temperature changes, we feel that capturing these alongside the smaller changes is critical to building a robust model. The underlying dynamics of the system, which we hope the regression model is able to learn, drives the full range of temperature changes seen. As such if we build a regressor which is unable to capture the extreme levels of change, this would indicate the model is not fully learning the physical dynamics as was intended. Capturing these extremes is also critical to obtaining a model which could (with further development) lead to a feasible alternative forecast system. ~~While these extremes are limited in their occurrence, their impact on the ocean system is notable, particularly if we were interested in longer prediction timescales. A model unable to capture these fails to provide a useful starting point for development of an ocean forecast system. Although we note that the development of a data-driven forecast system is not the focus of this work, the ability of the model developed here to capture extremes is to some extent relevant from that perspective.~~

Given the simplicity of the regressor used here, it is promising that it captures the extremes to the limited extent shown. However, the results also identify the need for

> more sophisticated methods which can better capture both the dominant dynamics, and the extreme cases.

We've also added the following to the conclusions,

~Line 513:

> That we see this behaviour in a simple model suggests that more complex models, capable of capturing the full higher-order non-linearity inherent in GCMs, are well placed to learn the underlying dynamics of these systems.
>
> The model developed here has a number of limitations, and a similar assessment of a more complex model, particularly one which can better capture the extreme behaviour alongside the more dominant dynamics would be of value to confirm this. The work carried out here uses a very idealised and coarse resolution simulator to create the dataset used for training and validation. Further investigation into how the complexity of the training data, and the resolution of the GCM used to create this dataset, impact the sensitivity of data-driven models would also be of further interest. Similarly, we assess model performance and model sensitivity over a single predictive step, but in forecasting applications data-driven models would most likely be used iteratively. Assessment of how model skill varies when iterating data-driven models has been carried out in the context of alternative data driven models. Looking alongside this to how the sensitivity of the model changes when using models iteratively would also provide further insight into this area.
>
> As data-driven models become competitive alternatives to physics driven GCMs, it is imperative to continue to investigate the sensitivity of these models, ensuring we have a good understanding of how these models are working and when it is valid to rely on them.

2. The authors quite correctly point out that their model cannot do autoregressive/ iterative prediction thus rendering their model not-useful to some extent. Still, interpretability of these models is a big step forward in this field. However, doing so from looking at predictions at a single time step may not be the right approach. There have been several studies that have shown that error propagation in this model is non-trivial and nonlinear. Thus, data-driven models that iteratively forecast the state of the atmosphere/ocean may show variable skill based on how far it directly forecasts and the error analysis may lead to starkly different results e.g., Figure 2 of this paper (https://agupubs.onlinelibrary.wiley.com/doi/full/10.1029/2020MS002203) shows the effect of changing the time step of prediction in the data-driven model and so does this paper (https://gmd.copernicus.org/preprints/gmd-2021-71/) . While it is definitely true that a physical variable that actually affects ocean temperature would probably lead to better skill if used as an input, this observation has been reported in the context of atmospheric dynamics in a more complex deep learning model (https://agupubs.onlinelibrary.wiley.com/doi/10.1029/2019MS00705) and is quite intuitive to be honest. It is still worth mentioning that the authors do a remarkable job at presenting these analyses which are quite comprehensive in the context of this problem.

As noted by the reviewer, the focus of our paper is in the sensitivity analysis of the model, rather than the long term predictive skill. When looking at the sensitivity of a model, the implications of looking over a single predictive step versus looking over an iterative forecast are notably different. When looking at the sensitivity of an iterated forecast (i.e. when applying the model many times to give a forecast multiple time-steps ahead), we see impacts from the way in which errors propagate

over iterations. This is interesting, and very important, particularly when considering this is in the context of meaningful predictive models; again however, this is not our focus here. We are especially interested in whether data-driven models can to some extent 'learn' the dynamics of the systems they are modelling rather than finding statistically valid, but not necessarily physically valid patterns. We think that this question is best addressed by firstly looking solely at single-step predictions, in order to avoid the propagation of errors over multiple step predictions confusing the results.

Again though, we note that there are many interesting questions around how errors propagate when iteratively forecasting with data-driven models and the sensitivity of models to this propagation. In particular considering the distinction between changes to forecast skill with forecast period and changes to forecast sensitivity with forecast period. It would make interesting further work to assess this question in the context of a model which can be iterated, as in the paper referenced here by the reviewer. We've added the following to the conclusions to highlight this for future work.

~Line 513:

> That we see this behaviour in a simple model suggests that more complex models, capable of capturing the full higher-order non-linearity inherent in GCMs, are well placed to learn the underlying dynamics of these systems.
> The model developed here has a number of limitations, and a similar assessment of a more complex model, particularly one which can better capture the extreme behaviour alongside the more dominant dynamics would be of value to confirm this. The work carried out here uses a very idealised and coarse resolution simulator to create the dataset used for training and validation. Further investigation into how the complexity of the training data, and the resolution of the GCM used to create this dataset, impact the sensitivity of data-driven models would also be of further interest. Similarly, we assess model performance and model sensitivity over a single predictive step, but in forecasting applications data-driven models would most likely be used iteratively. Assessment of how model skill varies when iterating data-driven models has been carried out in the context of alternative data driven models. Looking alongside this to how the sensitivity of the model changes when using models iteratively would also provide further insight into this area.
> As data-driven models become competitive alternatives to physics driven GCMs, it is imperative to continue to investigate the sensitivity of these models, ensuring we have a good understanding of how these models are working and when it is valid to rely on them.

I would like to emphasize that the analyses conducted in the study are very comprehensive and would had been much more impactful had it been done for a model that was useful for data driven prediction. Of course, with an increase in complexity of the model, these analyses would become non-trivial as well. However, at this current stage, the best validation accuracy of the data-driven model is 0.58 which questions the performance of the model especially for a single time step prediction. One of the reasons could be because of under-predicting the extremes which can be dealt in other ways as well.

Again we refer to our response to point 1 regarding the model skill. There are differing measures of model performance, and we think the RMS error is the most suitable here (and that the correlation coefficient value of 0.58 cannot straightforwardly be interpreted as 'low skill', especially in consideration of the high scores obtained when calculated both correlation coefficients and anomaly correlation coefficients for the predicted temperatures, rather than the

predicted increments.). We also emphasise again that our focus is not on forecasting extremes, but on first capturing the more common dynamics seen in the model.

Many thanks again for the comments and suggestions, and for the recognition of the comprehensive nature of this work.

---

## Author Comment (AC4)

**Reviewer 2**

Many thanks for your helpful comments and suggestions, and for taking the time to read and review our paper. We have responded to the comments in blue below (with the reviewers original comments in grey). Where we have made changes to the manuscript we've noted these below our comments, with red strikethrough text for deletions, green text for additions and black text indicating regions of no change.

Recently, the data-driven models have become a hot topic for the atmospheric and oceanic predictions for time scale from synoptic to interannual. Many cases indicate that such models have high predict skills and much less computational resource-consuming. However, as mentioned by the authors, some fundamental questions have no been answered yet, such as if the method can capture the predictability natural of the system and if we can physically explain the results. In the present study, a data-driven model has been developed for predicting the sea surface temperature in the short term. The method they used is a regression model with a non-linear term. The model has been trained using an idealized ocean model dataset as an observing system simulation experiment. They found that the model can predict the SST leading one day, and the dominant variables are also identified. After that, the sensitivity withholding experiments are conducted to identify critical physical variables and processes, like the vertical structure and the non-linear term. In general, this is an interesting and valuable paper and provides helpful information for this kind of data-driven model. Therefore, it is worth to published in GMD after the MINOR revision.

Just to note that the model predicts sea temperature at a range of depths - it is not intended to predict the change in SST, but rather it predicts the change in temperature at depth throughout the interior of the ocean.

Major questions:

1. I think the most critical problem is the resolution of the model is too coarse (2 degrees) compared with the predicting time scale (1 day). Because the movement of the ocean is much slower than that of the atmosphere, the surrounding points cannot affect the central point for one day. The only fast process is the convection due to instability. That is why the coefficient magnitude of the center point is much larger than other points in Figure 4. If the horizontal resolution is increasing, I guess that the results may also be changed because some processes may transport the signal of around points to the central point during one day. Therefore, I suggest the authors test the sensitivity of the resolution further.

Firstly, we stress that figure 4 does not show the impact of the central point alone. The top panel (figure 4a) is the impact averaged over all input points (the central point and all neighbouring point), and the bottom panel (figure 4b) shows the interaction between pairs of input points. Assuming the bright banding in figure 4b is what is being referred to here, this shows the polynomial terms are heavily weighted which combine information from the central point and surrounding points. This reflects the importance of *interaction* between the temperature at the central point and its neighbours, rather than the importance of the central point itself. We would expect this near-neighbour interaction to remain a dominant feature at all resolutions. We've amended the caption for figure 4 to clarify this;

Figure 4 caption:

> Coefficients of the control regressor. Top: Coefficients averaged over all input locations for each variable type, and each set of non-linear combinations of variables. Bottom:  Coefficients for

polynomial terms representing Temperature-Temperature interactions across all pairs of input locations.

Regarding testing the sensitivity of the resolution, thanks for this suggestion - it is certainly an interesting question as to how the resolution would change the ability of data-driven methods to learn the dynamics of the ocean and the sensitivity of the learned equations.

To a large extent these questions around the interaction between the spatial resolution used and the predicting time scale would be equally well considered by changing the temporal resolution - something we have considered in response to point 5, please see below.

Unfortunately assessing the sensitivity of the model to varying spatial resolution would require significant effort (it would involve re-running multiple simulations of the underlying MITgcm simulator, training the full set of new regressors on this data, and then analysing these results). It would also take considerable computational resource in comparison to the existing work - even doubling the resolution would take 10 times the compute resource running the simulator and would result in around 4 times as much data, which would in turn also lead to additional compute resource needed to train and validate the regressor.

Whilst acknowledging this investigation would make for interesting further work, we do not feel this is in the scope of this paper. We've added the following to the manuscript to reflect the potential of this for further work.

~Line 513:

> That we see this behaviour in a simple model suggests that more complex models, capable of capturing the full higher-order non-linearity inherent in GCMs, are well placed to learn the underlying dynamics of these systems.
> The model developed here has a number of limitations, and a similar assessment of a more complex model, particularly one which can better capture the extreme behaviour alongside the more dominant dynamics would be of value to confirm this. The work carried out here uses a very idealised and coarse resolution simulator to create the dataset used for training and validation. Further investigation into how the complexity of the training data, and the resolution of the GCM used to create this dataset, impact the sensitivity of data-driven models would also be of further interest. Similarly, we assess model performance and model sensitivity over a single predictive step, but in forecasting applications data-driven models would most likely be used iteratively. Assessment of how model skill varies when iterating data-driven models has been carried out in the context of alternative data driven models. Looking alongside this to how the sensitivity of the model changes when using models iteratively would also provide further insight into this area.
> As data-driven models become competitive alternatives to physics driven GCMs, it is imperative to continue to investigate the sensitivity of these models, ensuring we have a good understanding of how these models are working and when it is valid to rely on them.

2. In the withholding experiments, we can find that the errors are smaller than the control experiments in some places, such as withholding the information about the vertical structure in Figure 6 and withholding currents in Figure 7. How to understand these results? Is there some information that will bring negative effects or significant errors into the model? Can we find an optimal combination of all this information?

We have made some changes to the manuscript to further clarify this point. We hope this clarifies things. It is not the case that the additional information brings errors to the model, but that the best

*overall* equation for the control is not better at every individual prediction, compared to the best *overall* equation developed in each of the withholding experiments.

~Line 408:

Interestingly this regressor shows some regions (the deep water in the south of the domain) where the errors are notably improved in a regressor using only 2-d information.  In this work we have developed a regressor which learns one equation to be applied across all grid boxes in the domain. We optimise for best performance averaged over all relevant grid cells, but this does not enforce the best possible performance over each individual grid point/region, and so some versions of the model will favour certain types of dynamics than others. Given this, it is not unexpected that the new equations discovered for the withholding experiments (which also optimise for best performance averaged over the entire domain interior), may outperform the control in some locations, despite being poorer overall. Here we see that the control model is able to perform well across the domain, and optimises for good performance overall (see ~f~Fig. 3b), rather than the much more varied performance seen in the withholding experiments (Fig. 6b). It seems that as the model which withholds vertical information is not capable of performing well in many regions of the domain, a solution is found which highly optimises performance in other regions to minimise error overall. This highlights the limitations of our method, and the need for more complex data-driven models which can better adjust to the wide variety of dynamics shown across the domain. It would be possible to produce a plethora of simple regression models, each of which are optimised for different locations within the domain, and combine these to produce a domain wide prediction. However, this would be a far more computationally demanding challenge, and would bring with it large risks of overfitting. With this sort of design, each regional model, seeing only a subset of dynamics, would be less likely to 'learn' the underlying dynamics of the ocean, and instead learn statistically accurate but dynamically less valid local patterns. However, other more sophisticated methods could be explored however to find a single model which has the complexity to better capture the detailed non-linear dynamics in the ocean.

~Line 447:

Figure 7 shows the spatially averaged errors from this regressor along with the difference between these and the errors from the control model. Again we see a small number of points where errors are reduced with the simplified models. This is for the same reasons as described in Sect 4.3.2

3. The configurations of the model or experiment are not present very clear. Please give more information about the experiment in the manuscript, such as the vertical levels, the time scale of the restoring, the observation of the restoring sea surface temperature and salinity, and the coefficients of the GM scheme.

Thanks for this suggestion. The underlying MITgcm simulation is published in Munday et al (2013)[1], but for reader ease we have expanded the description of the configuration within the
* * *
[1] Munday, D. R., Johnson, H. L., Marshall, D. P., Munday, D. R., Johnson, H. L., and Marshall, D. P.: Eddy Saturation of Equilibrated Circumpolar Currents, Journal of Physical Oceanography, 43, 507–532, https://doi.org/10.1175/JPO-D-12-095.1, http://journals.ametsoc.org/doi/abs/10.1175/JPO-D-12-095.1, 2013

paper, including adding a table with key parameters, and pointed the reader more clearly to the aforementioned paper.

~Line 95:

> This configuration, whilst relatively simple, captures the fundamental dynamics of the ocean, including a realistic overturning circulation. The configuration is briefly described here, with key parameters given in Table 1. For further details the reader is referred to Munday et al. (2013).
>
> The domain runs from -60ºS to 60ºN, and is just over 20º wide in longitude. The domain is bounded by land along the northern (and southern) edge and a strip of land runs along the eastern (and western) boundary from 60ºN to 40ºS (see Fig.1a). Below this, in the southern-most 20º, the simulator has a periodic boundary condition, allowing flow which exits to the east (west) to return to the domain at the western (eastern) boundary. The domain has flat-bottom bathymetry of 5000m over most of the domain, with a 2º region of 2500m depth at the southern-most 20º of the eastern edge (i.e. the spit of land forming the eastern boundary continues to the southern boundary as a 2500m high sub surface ridge)
>
> The simulator is run with 42 (unevenly spaced) depth levels, following a Z-coordinate, with the surface layer being the thinnest at 10m, and the bottom 10 levels being the maximum at 250m , and has There are 11 cells in the X (longitudinal) direction, and 78 cells in the Y (latitudinal) direction. The grid spacing is 2º in the Y direction, with the X spacing scaled by the cosine of latitude to maintain approximately square grid boxes (this means grid cells close to the poles are about a factor of 4 smaller in area than those near the equator, but all cells remain approximately square). The simulator has a 12 hour time step (two steps per day), with fields output daily. We focus on daily-mean outputs, rather than the instantaneous state.
>
> At 2º resolution the simulator is not eddy resolving, but uses the Gent–McWilliams parameterisation (Gent and Mcwilliams, 1990) to represent the effects of ocean eddy transport. We ran the simulator with a strong surface restoring condition (see Table 1) on Temperature and Salinity — thus fixing the surface density. We apply simple jet-like wind forcing, constant in time, with a sinusoidal distribution (see Table 1) between 60ºS and 30ºS, with a peak wind stress value of 0.2 Nm$^{-2}$ at 45ºS.
>
> Table 1:

**Table 1.** Key parameter information for MITgcm simulation

| Parameter | Value |
|---|---|
| Grid spacing (horizontal) | $2°$ |
| Vertical levels | 42 unevenly spaced vertical levels ranging from 10m to 250m |
| Harmonic viscosity (momentum) | $0.0075 m^4 s^{-1}$ |
| vertical viscosity (momentum) | $10^{-3} m^2 s^{-1}$ |
| GM coefficient | $1000 m^4 s^{-1}$ |
| Reference diapycnal diffusivity | $3\mathrm{e}^{-5} m^2 s^{-1}$ |
| Wind stress | $0.2 sin^2[\pi(\theta+60)/30] Nm^{-2}$ for $-60 < \theta < -30$ |
| Restoring time scale for salinity | 30 days |
| Restoring salinity | $34. + 3/2([1+cos(\pi\theta/240)] PSU$ |
| Restoring time scale for potential temperature | 10 days |
| Restoring potential temperature | $30 sin[\pi(\theta+60)/120]°C$ for $theta < 0$
 $5 + 25 sin[\pi(\theta+60)/120]°C$ for $theta > 0$ |

4. The method of selecting data is also not present very clear. For instance, I do not really understand how to choose the data every 200-day and deal with the 3D variables at the surface. Please say something more about the details.

Thank you for highlighting the need for more explanation here. We have made the following changes to the text to clarify these points.

~Line 136:

 The data is highly auto-correlated, i.e. fields are similar, particularly when considering fields which are temporally close. This strong auto-correlation, found in many weather and climate applications, impacts the ability of the algorithm. Therefore, as is common practice, we sub-sample in time to remove some of the co-dependent nature of the training data, better optimising the ability of the data-driven method. There are also computational constraints limiting the total size of our dataset. This leads us to choose a subsampling rate of 200 days, so every 200th field from the simulator is used in the dataset, and the rest discarded. This provides a balance between having large datasets (which in general benefit the algorithm), whilst also fitting within computational constraints, and limiting auto-correlation within the dataset.

~Line 140:

 For every 200th day, we take all grid points from the model interior. We exclude points next to land and points at the surface and seabed, as the algorithm developed here is not suitable for forecasting these points --- the regressor requires input from surrounding points, and so is only suitable for predicting the interior of the domain.

5. In the present study, the authors only show the results of one-day prediction. I am curious how the model performs when the predicted time scale becomes longer, like 5-day or 10-day. I suggest the authors further discuss the skill of the model for more extended timescale prediction.

Many thanks for this suggestion. We have added a short appendix (Appendix C in the manuscript) showing how the model performs when predicting over a 5-day, 10-day and 20-day period, and a brief discussion of these results.

**Appendix C: Predicting over longer timescales**

**Table 2.** Table showing RMS errors, skill scores and correlation coefficients for 4 models trained to predict temperature change over increasing forecast period. RMS errors increase with forecast period, but skill scores and correlation coefficients are largely unaffected.

| | RMS error (°C) | | Skill score ($1 - \frac{modelRMS}{PersistenceRMS}$) | | Correlation coefficients | |
|---|---|---|---|---|---|---|
| | Training | Validation | Training | Validation | Training | Validation |
| Control (1 day forecast step) | 5.61e-5 | 9.89e-5 | .45 | .14 | .84 | .58 |
| Persistence over 1 day forecast step | 1.02e-4 | 1.15e-4 | - | - | - | - |
| 5 day forecast step | 2.79e-4 | 4.72e-4 | .45 | .14 | .83 | .59 |
| Persistence over 5 day forecast step | 5.07e-4 | 5.49e-4 | - | - | - | - |
| 10 day forecast step | 5.56e-4 | 9.27e-4 | .45 | .14 | .83 | .60 |
| Persistence over 10 day forecast step | 1.00e-3 | 1.08e-3 | - | - | - | - |
| 20 day forecast step | 1.07e-3 | 1.83e-3 | .45 | .14 | .84 | .60 |
| Persistence over 20 day forecast step | 1.95e-3 | 2.13e-3 | - | - | - | - |

We ran three additional versions of the regression model predicting 5, 10 and 20 days ahead, and compared the results with the regressor predicting a single day forecast day ahead. To clarify, this was not based on using the regressor iteratively, as the regressor is not designed to be used in this way. Instead the regressor makes a single forecast step of 5, 10 or 20 days, in place of the 1-day forecast step used in the control and throughout the paper. We consider the effect this has on the predictions. Table C1 shows the RMS error, skill score and correlation coefficients for the regressors trained

to predict 1, 5, 10 and 20 days ahead, along with the RMS errors for persistence forecasts over the same forecast length.

We can see that the correlation coefficient changes very little between the regressors. Correlation coefficients indicate how well the regressor captures the 'pattern' of the data, and at all forecast lengths the regressors do well at capturing these.

Generally RMS error is a more useful statistic in forecasting problems, as it gives an indication of average error per prediction. We can see that the RMS is larger with longer forecast lengths, over both the training and validation sets, meaning predictions have greater error over longer forecast lengths. This is to be expected, as predicting further ahead is a more challenging task. Temperature changes are larger over longer time periods and the dynamics of the underlying simulator (and the real ocean) mean that the temperature change at a particular point over a longer time period is driven by points increasingly further away, and in increasingly non-linear ways. As we only provide the regressor with information from directly neighbouring points as inputs, when looking at temperature changes over longer time periods, when points further away influence temperature change, the regressor is increasingly limited by the lack of input information. Similarly as the regressor is only able to represent a small amount of non-linearity, we would expect predicting further ahead to become more challenging.

We also consider how much of this increased error is related to the problem becoming harder with longer forecast step, or if there is any indication that the regression model is inherently unsuitable for forecasting over these longer forecast steps. By incorporating the baseline persistence RMS error, which also increases as the problem becomes harder, the skill score gives an indication of this differentiation. We see the skill scores remain constant (to two significant figures) regardless of the length of forecast step. This shows that while the model RMS error increases, this is likely to be due to the increasing difficulty of the prediction problem, and not a sign that the model itself is unsuited to predicting across these longer timescales.

This is a particularly interesting result in the context of data-driven forecasting. Traditional GCMs, such as the MITgcm simulator used to create the training and validation datasets, are limited in the length of forecast step that can be taken due to numerical constraints. For the configuration shown here however, we obtain similar skill even when forecasting over far larger steps than would be possible in the simulator, making this type of model far more efficient. These results warrant further investigation, in particular to see if similar patterns are shown with more complex configurations, and if the sensitivity of the regressor changes with increasing forecast length.